# Non-Equilibrium Liouville and Wigner Equations: Classical Statistical Mechanics and Chemical Reactions for Long Times

**DOI:** 10.3390/e21020179

**Published:** 2019-02-14

**Authors:** Ramon F. Álvarez-Estrada

**Affiliations:** Departamento de Fisica Teorica, Facultad de Ciencias Fisicas, Universidad Complutense, 28040 Madrid, Spain; rfa@ucm.es; Tel.: +34-91-3944-595

**Keywords:** non-equilibrium Liouville and Wigner distributions, equilibrium solutions and orthogonal polynomials, long-term irreversible approach of non-equilibrium moments to thermal equilibrium, chemical reactions for two and three particles, 05.20.-y, 05.10.Gg, 05.20.Jj, 05.30.-d

## Abstract

We review and improve previous work on non-equilibrium classical and quantum statistical systems, subject to potentials, without ab initio dissipation. We treat classical closed three-dimensional many-particle interacting systems without any “heat bath” (hb), evolving through the Liouville equation for the non-equilibrium classical distribution $W_{c}$, with initial states describing thermal equilibrium at large distances but non-equilibrium at finite distances. We use Boltzmann’s Gaussian classical equilibrium distribution $W_{c,eq}$, as weight function to generate orthogonal polynomials ($H_{n}$’s) in momenta. The moments of $W_{c}$, implied by the $H_{n}$’s, fulfill a non-equilibrium hierarchy. Under long-term approximations, the lowest moment dominates the evolution towards thermal equilibrium. A non-increasing Liapunov function characterizes the long-term evolution towards equilibrium. Non-equilibrium chemical reactions involving two and three particles in a hb are studied classically and quantum-mechanically (by using Wigner functions *W*). Difficulties related to the non-positivity of *W* are bypassed. Equilibrium Wigner functions $W_{eq}$ generate orthogonal polynomials, which yield non-equilibrium moments of *W* and hierarchies. In regimes typical of chemical reactions (short thermal wavelength and long times), non-equilibrium hierarchies yield approximate Smoluchowski-like equations displaying dissipation and quantum effects. The study of three-particle chemical reactions is new.

## 1. Introduction

Research leading from equilibrium statistical mechanics [1,2,3,4,5,6] to non-equilibrium statistical mechanics [2,7,8,9] in both classical and quantum regimes faces many open fundamental difficulties.

One crucial hallmark is that the off-equilibrium evolution of statistical systems displays stochasticity: see [10,11,12] in the classical regime and [13,14,15,16,17,18,19] in the quantum one, specifically, in the framework of the theory of open quantum systems. For complementary background in classical (thermodynamical) frameworks, see [20,21,22].

Chemical reactions constitute a very important and wide class of phenomena in which equilibrium and non-equilibrium statistical mechanics play essential roles: such phenomena are genuinely quantum-mechanical, although classical approaches can provide useful information under various approximations. See [5,6] and references therein for studies devoted to the equilibrium chemical constant. Much research has also been devoted to non-equilibrium chemical reactions: see in particular [11,20,22], for stochastic and thermodynamical approaches. The non-equilibrium rate constant is treated in [11] by means of a thorough analysis of the Kramers equation.

Use has been made, by previous authors, of infinite hierarchies in the analysis of several stochastic equations [23,24]. The present author has carried out several detailed analyses of the classical Liouville and quantum Wigner equation, through infinite hierarchies for suitable non-equilibrium moments, in order to analyze approaches to thermal equilibrium in long-term approximations [25,26,27,28,29,30,31,32]. Motivated by previous work on non-equilibrium DNA thermal denaturation (the dynamics of two interacting classical three-dimensional macromolecular chains) [33], binary chemical reactions have been studied quantum-mechanically, based on the non-equilibrium Wigner function [34].

The intention of this work is to review and extend the work mentioned above in [25,26,27,28,29,30,31,32] and in [34]. To outline, we shall focus on the dynamics of non-equilibrium non-relativistic systems implied by the classical Liouville equation and the quantum Wigner one [2]. Throughout our study, we shall exclude by assumption, from the outset and in all cases, ab initio dissipation; there will be vanishing friction due to any other system. We shall omit throughout any detailed analysis of the states of the “heat bath” hb (just retaining the fact that it establishes the temperature) and, in an effective way, focus exclusively on the states of the systems under study. The corresponding equilibrium distributions will be used to generate an infinite family of orthogonal polynomials. In turn, the latter will allow construction of non-equilibrium moments which, through either the Liouville equation or the Wigner one, will imply infinite linear hierarchies. These hierarchies, under suitable assumptions and approximations, will yield an approach to thermal equilibrium for the long term. The formalism will enable a theoretical analysis of chemical reactions.

The above program faces various difficulties, if it is to be treated successively. We shall begin with simpler one-dimensional settings (namely, one particle subject to an external potential and to a hb at thermal equilibrium) in both classical (Section 2) and quantum (Section 4) cases, to illustrate how difficulties will be bypassed. Section 3 deals with classical closed three-dimensional many-particle interacting systems without any hb, evolving through the Liouville equation for the non-equilibrium classical probability distribution $W_{c}$ and with initial states describing thermal equilibrium at large distances but non-equilibrium at finite ones. Section 5 outlines useful features of the non-equilibrium dynamics of N>1 quantum particles subject to a hb using Wigner functions, which will be useful in Section 6 and Section 7. Section 6 studies non-equilibrium chemical reactions involving two particles. Section 7 deals with a new short study of non-equilibrium chemical reactions involving three particles. Section 8 summarizes the contents of this article and distinguishes review material from new material, along with some comparative discussions. Some open problems are highlighted.

Throughout the text, successive assumptions, simplifications, and approximations are indicated separately. Successive results are explicitly indicated, for compactness, within the main text.

## 2. Open Classical One Particle Systems

This section will introduce techniques to be used directly in Section 3, dealing with many-particle systems.

### 2.1. One-Dimensional Case: Some General Aspects

Let a classical particle, with mass *m*, position *x* and momentum *q*, be subject to a real potential V=V(x), in the presence of a “heat bath” (hb) at thermal equilibrium at temperature *T*. We shall employ the standard variable $\beta ={\left(k_{B}T\right)}^{-1}$ ($k_{B}$ being Boltzmann’s constant).

Assumptions: The potential is repulsive: V(x)≥0: either V(x)→0 as ∣x∣→+∞ or V(x)≡0.

The classical Hamiltonian of the particle is: $H_{c}=q^{2}/\left(2m\right)+V$. Let the classical particle be, at the initial time t=0, out of thermal equilibrium with the hb, and have a probability distribution function $W_{c,in}=W_{c,in}\left(x,q\right)\left(\geq 0\right)$ to be at the position *x* with momentum *q*. Then, the non-equilibrium particle could be, at time *t*(>0), at the position *x* with momentum *q*, with probability distribution function $W_{c}=W_{c}\left(x,q;t\right)\left(\geq 0\right)$. For instance, the hb could be air (at rest and at thermal equilibrium, at T≃300
${}^{\circ }$K) in a room, and the classical particle could be a virus or a grain of pollen, performing Brownian motion in air.

The time evolution of $W_{c}$ is given by the reversible Liouville equation:(1)$$\begin{array}{c} \frac{\partial W_{c}}{\partial t}+\frac{q}{m}\frac{\partial W_{c}}{\partial x}-\frac{\partial V}{\partial x}\frac{\partial W_{c}}{\partial q}=0 \end{array}$$with the initial condition $W_{c,in}$. The equilibrium (Boltzmann’s) distribution, the *t*-independent solution of Equation (Equation 1) describing thermal equilibrium of the particle with the hb, is: $W_{c}=W_{c,eq}=\exp\left[-\beta \left(q^{2}/\left(2m\right)+V\right)\right]$, Gaussian in *q*.

Let $q_{eq}={\left[2m/\beta \right]}^{1/2}$ be the physically natural fixed (scaling) momentum determined by *T*. We shall define $y=q/q_{eq}$. We shall introduce the denumerably infinite family of all (unnormalized) polynomials in *y*: $H_{n}=H_{n}\left(y\right)$ (n=0,1,2,3,…), the standard *n*-th Hermite polynomial [35], orthogonalized in *y* (for fixed *x*) by using $W_{c,eq}$ as (Gaussian) weight function. With $H_{0}\left(y\right)=1$, one has, for $n\neq n^{\prime }$ and any *x* (left unintegrated):(2)$$\begin{array}{c} \int_{-\infty }^{+\infty }dyW_{c,eq}\left(x,q\right)H_{n}\left(y\right)H_{n^{\prime }}\left(y\right)=0 \end{array}$$

The orthonormalized polynomials are $H_{n}\left(y\right)/{\left(\pi^{1/2}2^{n}n!\right)}^{1/2}$. We shall introduce the (normalized) non-equilibrium classical moments $W_{c,n}=W_{c,n}\left(x;t\right)$ (n=0,1,2,…) of $W_{c}$ [10,23,25,26]:(3)$$\begin{array}{c} W_{c,n}=W_{c,n}\left(x,t\right)=\int d\left(q/q_{eq}\right)\frac{H_{n}\left(q/q_{eq}\right)}{{\left(\pi^{1/2}2^{n}n!\right)}^{1/2}}W_{c}\left(x,q;t\right) \end{array}$$$W_{c,0}$ is the marginal probability distribution for *x*. If $W_{c}=W_{c,eq}$, then $W_{c,eq,0}=\pi^{1/4}\exp\left[-\beta V\right]$ and $W_{c,eq,n}=0$, n=1,2,… Equation (Equation 3) can also be applied to the initial off-equilibrium distribution $W_{c,in}$ and gives the initial moments, $W_{c,in,n}$. One has the following (formal) expansion for $W_{c}$:(4)$$\begin{array}{c} W_{c}=W_{c,eq}\left(x,q\right)\sum_{n=0}^{+\infty }W_{c,n}\left(x;t\right)\frac{H_{n}\left(y\right)}{{\left(\pi^{1/2}2^{n}n!\right)}^{1/2}} \end{array}$$

### 2.2. 3-Term Hierarchy and Operator-Continued Fractions

Equations (Equation 1)–(Equation 3) yield an exact three-term non-equilibrium hierarchy for all $W_{c,n}$’s. It is convenient to work with the symmetrized moments $g_{n}=g_{n}\left(x,t\right)=W_{c,eq,0}{\left(x\right)}^{-1/2}W_{c,n}\left(x,t\right)$. The hierarchy for the $W_{c,n}$’s becomes an exact three-term hierarchy for the $g_{n}$’s, for any *n*, as a result: (5)$$\begin{array}{c} \frac{\partial g_{n}}{\partial t}=-M_{c,n,n+1}g_{n+1}-M_{c,n,n-1}g_{n-1} \end{array}$$(6)$$\begin{array}{c} M_{c,n,n\pm 1}g_{n\pm 1}\equiv {\left[\frac{\left(n+\left(1/2\right)\left(1\pm 1\right)\right)k_{B}T}{m}\right]}^{1/2}\left[\frac{\partial g_{n\pm 1}}{\partial x}-\frac{\left(\pm 1\right)g_{n\pm 1}}{2k_{B}T}\frac{\partial V}{\partial x}\right] \end{array}$$with initial condition $g_{in,n}=W_{c,in,n}/W_{c,eq,0}^{1/2}$. One key fact is that $M_{c,n,n+1}$ and $-M_{c,n+1,n}$ are the adjoint of each other (Equations (Equation 5) and (Equation 6) providing an anti-Hermitian hierarchy).

Let us consider the Laplace transforms ${\tilde{g}}_{n}={\tilde{g}}_{n}\left(s\right)=\int_{0}^{+\infty }dtg_{n}\exp\left(-st\right)$, with inverse $g_{n}=\int_{c-i\infty }^{c+i\infty }\left(ds/2\pi i\right){\tilde{g}}_{n}\exp\left(st\right)$ (*c* being real and such that ${\tilde{g}}_{n}\left(s\right)$ is analytic in the half-plane Res>c of the complex *s*-plane). This definition and Equation (Equation 5) yield the three-term hierarchy for ${\tilde{g}}_{n}$:(7)$$\begin{array}{c} s{\tilde{g}}_{n}=g_{in,n}-M_{c,n,n+1}{\tilde{g}}_{n+1}-M_{c,n,n-1}{\tilde{g}}_{n-1}\; \end{array}$$

The hierarchy Equation (Equation 7) can be solved formally by extending to it standard procedures for solving numerical three-term linear recurrence relations in terms of continued fractions (see, for instance, [10]). That formal procedure yields all ${\tilde{g}}_{n}\left(s\right)$, for any n=1,…, in terms of sums of products of certain *s*-dependent linear operators (actually, integral operators) $D[n^{\prime };s]$, $n^{\prime }\geq n$, acting upon ${\tilde{g}}_{n-1}\left(s\right)$ and upon all $g_{in,n^{\prime }}$’s, with $n^{\prime }\geq n$. The linear operators D[n;s]’s are defined recurrently through:(8)$$\begin{array}{c} D\left[n;s\right]={\left[sI-M_{c,n,n+1}D\left[n+1;s\right]M_{c,n+1,n}\right]}^{-1} \end{array}$$
*I* is the unit operator. By iteration of Equation (Equation 8), D[n;s] becomes a formal infinite continued fraction of products of the linear operators $M_{c,n,n+1}$ and $M_{c,n+1,n}$ (which do not commute, in general). That formal infinite continued fraction of operators reads:(9)$$\begin{array}{c} D\left[n;s\right]=\frac{I}{sI-M_{c,n,n+1}\frac{I}{sI-M_{c,n+1,n+2}\frac{I}{sI-M_{c,n+2,n+3}\frac{I}{sI-\ldots }M_{c,n+3,n+2}}M_{c,n+2,n+1}}M_{c,n+1,n}} \end{array}$$

Simplification—for a simpler hierarchy and a clearer exposition, without loss of generality, let us assume that $W_{c.in,n^{\prime }}=0$ for $n^{\prime }\geq 1$, with $W_{c,in,0}\neq 0$.

By solving Equation (Equation 7) for n(≥1) (through Equation (Equation 8)), we get the following result, without approximations, so far: (10)$$\begin{array}{c} s{\tilde{g}}_{0}=W_{c,eq,0}^{-1/2}W_{c,in,0}-M_{c,0,1}{\tilde{g}}_{1} \end{array}$$(11)$$\begin{array}{c} {\tilde{g}}_{n}\left(s\right)=-D\left[n;s\right]M_{c,n,n-1}{\tilde{g}}_{n-1}\left(s\right),\;n\geq 1 \end{array}$$

### 2.3. Properties of D[n;s]


Let us choose n(≥1) and fix s=ϵ>0 (real and suitably small) in any D[n;s]. Then, the following crucial properties appear to hold (in general, for either V≠0 or V=0) [25,26]: if D[n+1;ϵ] were Hermitian and if all its eigenvalues (which would be real) were non-negative, then the same would hold true for D[n;ϵ]. It is easy to confirm the validity of that property if D[n+1;ϵ] is replaced by a 2×2 matrix with the above (non-negativity and Hermiticity) properties and $M_{n,n+1}$ and $M_{n+1,n}$ are chosen as 2×2 matrices such that $M_{n,n+1}$ be the adjoint of $-M_{n+1,n}$. In particular, a confirming example with a 2×2 matrix is given in Appendix D in [26]. Then, through iterative arguments, as a result, D[n;ϵ]’s, n=0,1,2,3…, turn out to be Hermitian, and all their eigenvalues are non-negative, for V≠0, in general.

To understand the structure and convergence of the operator-continued fractions D[n;s], we shall take V≡0 and let ϵ>0 first, so as to allow for ϵ→0 later [25]. Let us perform a spatial Fourier transformation from configuration space (*x*) to wavevector space (*k*), by applying ${\left(2\pi \right)}^{-1/2}\int dx\exp\left(-ikx\right)$. Let $e\left(k\right)\equiv {\left(2m\right)}^{-1}k_{B}Tk^{2}$. Then, the Fourier transform of the operator-continued fraction in Equation (Equation 8), for Res>0 and n≥0 is the following continued fraction involving ordinary functions $D_{1}\left[k;n;s\right]$:(12)$$\begin{array}{c} D_{1}\left[k;n;s\right]=[s+2e\left(k\right)\left(n+1\right)D_{1}{\left[k;n+1;s\right]}^{-1} \end{array}$$

By iteration, $D_{1}\left[k;n;s\right]$ becomes the following ordinary continued fraction (involving no non-commuting operators):(13)$$\begin{array}{c} D_{1}\left[k;n;s\right]=\frac{1}{s+2e\left(k\right)\left(n+1\right)\frac{1}{s+2e\left(k\right)\left(n+2\right)\frac{1}{s+2e\left(k\right)\left(n+3\right)\frac{1}{s+2e\left(k\right)\left(n+4\right)\frac{1}{s+\ldots \ldots ..}}}}} \end{array}$$to be compared to Equation (Equation 9) (involving non-commuting operators).

It will be more convenient to make use of (shorter) standard notations [36] for ordinary infinite continued fractions (involving only commuting quantities). Accordingly, Equation (Equation 13) reads, in standard notations: (14)$$\begin{array}{c} D_{1}\left[k;n;s\right]=\frac{1}{s+}\frac{2e(k)(n+1)}{s+}\frac{2e(k)(n+2)}{s+}\ldots =\frac{1}{e{\left(k\right)}^{1/2}}.\left[\frac{2^{-1}}{z+}\frac{2^{-1}\left(n+1\right)}{z+}\frac{2^{-1}\left(n+2\right)}{z+}\ldots \right]. \end{array}$$with $z=s/(2e{\left(k\right)}^{1/2})$. On the other hand, the continued fraction can be expressed as:(15)$$\begin{array}{c} \frac{2^{-1}}{z+}\frac{2^{-1}\left(n+1\right)}{z+}\frac{2^{-1}\left(n+2\right)}{z+}\ldots =\frac{i^{n}erfc\left(z\right)}{i^{n-1}erfc\left(z\right)}\;. \end{array}$$
$i^{n}erfc\left(z\right)$ being the *n*-th repeated integral of the complementary error function ($i^{0}erfc\left(z\right)=erfc\left(z\right)$) [36]. As *z* varies in 0≤z<+∞, the values given by Equation (Equation 15) vary in a finite interval. With ϵ>0 and by using [36], one gets: $D_{1}\left[k;1;\epsilon \right]\to \epsilon^{-1}$ as k→0, while $D_{1}\left[k;1;\epsilon \right]\to {\left(\pi e\left(k\right)\right)}^{-1/2}$ as ∣k∣→∞. For s=ϵ=0, the behavior of $D_{1}$ is different. Equation (Equation 14) gives:(16)$$\begin{array}{c} D_{1}\left[k;n;s=\epsilon =0\right]=D_{1}\left[k;n;0\right]={\left[2e{\left(k\right)}^{1/2}\right]}^{-1}\frac{1}{0+}\frac{2^{-1}\left(n+1\right)}{0+}\frac{2^{-1}\left(n+2\right)}{0+}\ldots \end{array}$$

The continued fraction in Equation (Equation 16) can, in turn, be evaluated in terms of the standard Gamma function Γ [36]:(17)$$\begin{array}{c} \frac{2^{-1}}{0+}\frac{2^{-1}\left(n+1\right)}{0+}\frac{2^{-1}\left(n+2\right)}{0+}\ldots \;.=\frac{\Gamma ((n/2)+1/2)}{2\Gamma ((n/2)+1)} \end{array}$$

The ratio in Equation (Equation 17) behaves as $n^{-1/2}$, as n→+∞. We state that the various classical operator-continued fractions will also behave, on average, as $n^{-1/2}$ for large *n*: we shall omit details [25]. For k→0, $D_{1}\left[k;n;0\right]$ diverges as $k^{-1}$ (due to $e{\left(k\right)}^{-1/2}$). Then, $\int dkD_{1}\left[k;n;0\right]$ also diverges near k=0.

### 2.4. An Approximate Ansatz for Equation (Equation 8), for Large n

Using the developments in Section 2.3 as insights, we shall now outline an approximate ansatz, in order to simplify Equation (Equation 8) under the following conditions.

Approximation—Let s=ϵ>0 be real and suitably small and $n_{0}^{\prime }$ be a suitably large positive integer. Then, for $n,n^{\prime },n^{\prime \prime }>n_{0}^{\prime }$ and as an approximation, we disregard the non-commutativities of all $M_{c,n,n+1}$, $M_{c,n+1,n}$ and $D[n^{\prime \prime };\epsilon ]$ among themselves and regard all of them as commuting operators.

Notice also that the non-negative Hermitean operator $\left(-M_{c,n,n+1}M_{c,n+1,n}\right)/\left(n+1\right)$ is independent of *n* and that the spectrum of all eigenvalues of $\left(-M_{c,n,n+1}M_{c,n+1,n}\right)$ varies in (0,+∞). Then, after a direct formal manipulation inspired on the structures of Equations (Equation 14) and (Equation 16), Equation (Equation 8) becomes:(18)$$\begin{array}{c} D\left[n;\epsilon \right]={\left[2\left(-M_{c,n,n+1}M_{c,n+1,n}\right)/\left(n+1\right)\right]}^{-1/2}\frac{1}{z_{1}+}\frac{2^{-1}\left(n+1\right)}{z_{1}+}\frac{2^{-1}\left(n+2\right)}{z_{1}+}\ldots ]. \end{array}$$with $z_{1}=\epsilon {\left[2\left(-M_{c,n,n+1}M_{c,n+1,n}\right)/\left(n+1\right)\right]}^{-1/2}$. A comparison with Equation (Equation 14) shows that Equation (Equation 18) is exact if V≡0.

Approximation—At this point, for suitably small ϵ, we assume that we can neglect the contribution of the non-negative Hermitean operator $z_{1}$.

By using Equations (Equation 16) and (Equation 17), we get the approximate (ϵ-independent) operator, as a result:(19)$$\begin{array}{c} D\left[n;\epsilon \right]={\left[2\left(-M_{c,n,n+1}M_{c,n+1,n}\right)/\left(n+1\right)\right]}^{-1/2}\frac{\Gamma ((n/2)+1/2)}{\Gamma ((n/2)+1)} \end{array}$$

One expects that the dominant contribution to this approximate D[n;ϵ] is determined by the set of lowest (non-negative) eigenvalues of $\left(-M_{c,n,n+1}M_{c,n+1,n}\right)$.

### 2.5. Long-Term Approximation

Approximations—The long-term approximation for $n\geq n_{0}\geq 1$ reads in general (that is, without imposing $W_{c.in,n^{\prime }}=0$ for $n^{\prime }\geq 1$, with $W_{c,in,0}\neq 0$) as follows. One replaces for $n\geq n_{0}\geq 1$ any D[n;s] yielding ${\tilde{g}}_{n}$ in terms of ${\tilde{g}}_{n-1}$ by D[n;ϵ]: this approximation is not done for $n<n_{0}$, which will be crucial, and is the better, the larger $n_{0}$. We regard $D[n_{0};\epsilon ]$ for $n=n_{0}$ as a fixed (*s*-independent) operator. Some ansatz or approximation should be provided directly for $D[n_{0};\epsilon ]$, like the one in Section 2.4. For simplicity, let us continue with the simplification: $W_{c.in,n^{\prime }}=0$ for $n^{\prime }\geq 1$, with $W_{c,in,0}\neq 0$. Moreover, after the above long-term approximation, we shall continue with the same initial condition $W_{c,in,0}$ at t=0: it may well be that this amounts to another kind of approximation.

Then, for small *s*, we approximate for $n=n_{0}$ as: ${\tilde{g}}_{n_{0}}\left(s\right)≃-D\left[n_{0};\epsilon \right]M_{n_{0},n_{0}-1}{\tilde{g}}_{n_{0}-1}\left(s\right)$. The resulting hierarchy for ${\tilde{g}}_{n}$’s ($n=0,\ldots ,\;n_{0}-1$), through the inverse Laplace transform, yields a closed approximate irreversible hierarchy for $g_{n}$, $n=0,1,\ldots ,\;n_{0}-1$. The solutions of the last closed hierarchy for $g_{n}$ relax irreversibly, for large *t* and reasonable $W_{c,in,0}$, towards $W_{c,eq,0}^{1/2}\neq 0$ and $W_{c,eq,n}^{1/2}=0$, $n=1,\ldots ,\;n_{0}-1$ (thermal equilibrium) [25,26]. Then, for the long term, the dominant moment is $g_{0}$, while any $g_{n}$ with n>0 being the smaller, the larger *n* and t(>0) are (due to the behaviors of $M_{c,n,n\pm 1}$ and of D[n;ϵ] with *n*). Similar behaviors hold for $W_{c,0}$ and $W_{c,n}$ with n>0.

Simplification—We shall illustrate the above facts by taking, for simplicity, $n_{0}=1$.

Then, Equation (Equation 11) for small *s* and n=1 becomes: ${\tilde{g}}_{1}\left(s\right)≃-D\left[1;\epsilon \right]M_{c,1,0}{\tilde{g}}_{0}\left(s\right)$ (to be compared to Equation (Equation 11) before the approximation). Then, Equations (Equation 10) and (Equation 11) yield, by taking inverse Laplace transforms: (20)$$\begin{array}{c} \frac{\partial g_{0}}{\partial t}=-M_{c,0,1}g_{1} \end{array}$$(21)$$\begin{array}{c} g_{1}≃-D\left[1;\epsilon \right]M_{c,1,0}g_{0} \end{array}$$

One finds directly the irreversible Smoluchowski-like equation for the n=0 moment, as a result:(22)$$\begin{array}{c} \partial g_{0}/\partial t=M_{c,0,1}D\left[1;\epsilon \right]M_{c,1,0}g_{0} \end{array}$$with initial condition $W_{c,eq,0}^{-1/2}W_{c,in,0}$. 

The right-hand side of Equation (Equation 22) should be interpreted as

$\int_{-\infty }^{+\infty }dx^{\prime }\left(M_{c,0,1}D\left[1;\epsilon \right]M_{c,1,0}\right)\left(x,x^{\prime }\right)g_{0}\left(x^{\prime };t\right)$. Let: $\left(f_{1},f_{2}\right)=\int_{-\infty }^{+\infty }dxf_{1}{\left(x\right)}^{*}f_{2}\left(x\right)$ for suitable functions $f_{1}$ and $f_{2}$. Due to the Hermiticity of D[1;ϵ]: $\left(f_{1},M_{c,0,1}D\left[1;\epsilon \right]M_{c,1,0}f_{2}\right)=\left(M_{c,0,1}D\left[1;\epsilon \right]M_{c,1,0}f_{1},f_{2}\right)$, thereby checking that $M_{c,0,1}D\left[1;\epsilon \right]M_{c,1,0}$ is Hermitian. Moreover:

$\left(f_{1},M_{c,0,1}D\left[1;\epsilon \right]M_{c,1,0}f_{1}\right)=-\left(M_{c,1,0}f_{1},D\left[1;\epsilon \right]M_{c,1,0}f_{1}\right)\leq 0$ for arbitrary functions $f_{1}$, as all eigenvalues of D[1;ϵ] are ≥0. Let $f_{\lambda }\left(x\right)$ be an eigenfunction of the integral operator $M_{c,0,1}D\left[1;\epsilon \right]M_{c,1,0}$ with eigenvalue λ(≥0). Then, $\left(M_{c,0,1}D\left[1;\epsilon \right]M_{c,1,0}\right)\left(x,x^{\prime }\right)=\sum_{\lambda }\lambda f_{\lambda }\left(x\right)f_{\lambda }{\left(x^{\prime }\right)}^{*}$. $\sum_{\lambda }$ is a short-hand notation denoting integration and summation over the whole spectrum of $M_{c,0,1}D\left[1;\epsilon \right]M_{c,1,0}$. By expanding $W_{c,eq,0}^{-1/2}W_{c,in,0}=\sum_{\lambda }g_{in,\lambda }f_{\lambda }\left(x\right)$, with *x*-independent $g_{in,\lambda }$, the solution of (Equation 22) with the above initial condition is $g_{0}=\sum_{\lambda }g_{in,\lambda }f_{\lambda }\left(x\right)\exp\left(-\lambda t\right)$, which relaxes irreversibly as t→+∞ towards $g_{in,0}f_{0}\left(x\right)$, corresponding to λ=0. At equilibrium, one has: $g_{0}=W_{c,eq,0}^{1/2}$ (proportional to $f_{0}$), $M_{c,1,0}g_{0}=0$ and $g_{n}=0$, n=1,2,3…, consistently. Then, Equation (Equation 22) is (at least, with ϵ>0) as irreversible as the standard heat equation: for long t(>0) the dominant moment is $g_{0}$, while any $g_{n}$ with n>0 is negligible, $g_{n}$ being the smaller, the larger *n* and t(>0), and so on for the $W_{c,n}$’s.

Irreversible thermalization does not happen in the absence of long-term approximations. Then, the above approximations (D[n;s]≃D[n;ϵ] for $n\geq n_{0}$, but not for $n<n_{0}$), giving rise to the thermalization with the hb implemented in Equation (Equation 22), is an alternative way of introducing irreversibility out of the reversible Equation (Equation 1).

## 3. Closed Classical Many-Particle Systems: Long-Term Approximation and Arrow of Time

### 3.1. Initial State Motivated by Fluid Dynamics, Hierarchy, and Continued Fractions

We shall outline the main developments (details being given in [25]). We treat a closed large system of many (N≫1) classical non-relativistic particles, in three spatial dimensions, with spatial coordinates $x_{1}$,...,$x_{N}$ (≡(x)) and momenta $q_{1}$,...,$q_{N}$ (≡(q)). Let $x_{i,\alpha }$ and $q_{i,\alpha }$ be the Cartesian components of $x_{i}$ and $q_{i}$, respectively (i=1,…,N, α=1,2,3). The non-equilibrium classical probability distribution function is: $W_{c}=W_{c}\left(\left(x\right),\left(q\right);t\right)$.

Assumptions—(a) All particles, which are identical, have mass *m*. Neither a hb nor external friction mechanisms nor external forces are assumed. The physical (qualitative) idea is that the system contains a very large set ($s_{1}$) of degrees of freedom at large distances at thermal equilibrium at absolute temperature *T* with one another: $s_{1}$ plays the role of an (internal) hb. However, by assumption the system is in a non-equilibrium state, because it also contains a large set of degrees of freedom ($s_{2}$) at finite distances, which are off-equilibrium with the previous set $s_{1}$, and among themselves. We assume that the set $s_{1}$ is larger than the set $s_{2}$. Then, the initial non-equilibrium distribution $W_{c,in}=W_{c,in}\left(\left(x\right),\left(q\right)\right)$, which is known by assumption, describes thermal equilibrium at absolute temperature *T* for the set $s_{1}$ located at large distances, but off-equilibrium for the set $s_{2}$ located at finite distances [25]. The above qualitative idea will become clearer through $W_{c,in}$ and its properties, discussed below in (c).

(b) The interaction potential among the particles is: $V=\Sigma_{i,j=1,i<j}^{N}v_{ij}\left(|x_{i}-x_{j}|\right)$ and we suppose that all $v_{ij}\left(|x_{i}-x_{j}|\right)$ are repulsive (≥0) and tend quickly to zero for large $|x_{i}-x_{j}|$. $H_{c,N}={\left(2m\right)}^{-1}\Sigma_{i=1}^{N}\Sigma_{\alpha =1}^{3}q_{i,\alpha }^{2}+V$ is the classical *N*-particle Hamiltonian. Then, Boltzmann’s equilibrium (canonical) distribution at temperature *T* is: $W_{c,eq}=\exp\left[-\beta H_{c,N}\right]$.

(c) The initial distribution function $W_{c,in}$, at t=0, will be chosen to be: (i) qualitatively consistent with the idea [3,37], typical of Information Theory that one should employ only distribution functions compatible with the limited information available which, in turn, refers to expectation values of only a subset of observable dynamical variables (these ideas being imposed for t=0 only, but not for t>0); (ii) also consistent with standard variables employed in equilibrium statistical mechanics and fluid dynamics [2,7,8,9]. Then, our ansatz for $W_{c,in}$ will depend on a finite number (actually, 2+3=5) of functions of one single position vector, x: $\lambda_{k}=\lambda_{k}\left(x\right)$, k=0,2, and $\lambda_{1,\alpha }=\lambda_{1,\alpha }\left(x\right)$, α=1,2,3 (all of them being independent of time and on momenta). The expression for $W_{c,in}$ in terms of $\lambda_{k}$ and $\lambda_{1,\alpha }$ has appeared previously [2,9] (and is related to the Massieu-Planck function [9]). $W_{c,in}$, which describes thermal equilibrium with homogeneous temperature *T* for large distances (large ∣x∣) but non-equilibrium for intermediate and short distances ∣x∣ (with spatial inhomogeneities), reads:(23)$$\begin{array}{c} W_{c,in}={\left(N!\right)}^{-1}\exp[-\int d^{3}x(\lambda_{0}\left(x\right)\sum_{i=1}^{N}\delta^{\left(3\right)}\left(x_{i}-x\right)+\sum_{\alpha =1}^{3}\lambda_{1,\alpha }\left(x\right)\sum_{i=1}^{N}\frac{q_{i,\alpha }}{m}\\ \times \delta^{\left(3\right)}\left(x_{i}-x\right)+\lambda_{2}\left(x\right)\sum_{i=1}^{N}\left(\frac{q_{i}^{2}}{2m}+\frac{1}{2}\sum_{j=1,j\neq i}^{N}v_{ij}\left(|q_{i}-q_{j}|\right)\right)\delta^{\left(3\right)}\left(x_{i}-x\right)]\;. \end{array}$$$\delta^{\left(3\right)}$ denoting the three-dimensional Dirac delta function. Consistently with (i)–(ii) above, the λ’s will be uniquely determined in terms of 2+3=5
x-dependent observables (also independent of time and on momenta) typically employed in Fluid Dynamics, which, by assumption, are known at t=0: mass density, fluid velocity and some suitable energy density [9,25]. How does the equilibrium temperature *T* appear in this closed system, without hb? We accept that $\lambda_{2}\left(x\right)$ approaches quickly a non-vanishing constant, $\lambda_{2}\left(\infty \right)$, as ∣x∣ tends to *∞* along any direction and that the same holds for $\lambda_{0}\left(x\right)$. A similar statement holds for $\lambda_{1,\alpha }\left(x\right)$, the corresponding (large-∣x∣) limiting value being zero. At finite x, the off-equilibrium $\lambda_{2}\left(x\right)$, $\lambda_{0}\left(x\right)$ and $\lambda_{1,\alpha }\left(x\right)$ do depend on x and, so, differ from their respective constant (large-∣x∣) limiting values, which describe equilibrium. Then, consistency is achieved (*T* being thereby introduced) if, in the thermodynamical limit, $\lambda_{2}\left(\infty \right)$ tends to ${\left(k_{B}T\right)}^{-1}$(plus corrections which approach zero in that limit). The dominant contributions to various statistical averages at t=0 come from large x, up to corrections which decrease as *N* increases (and tend to vanish as N→∞). For a detailed analysis, see [25].

The reversible Liouville equation reads:(24)$$\begin{array}{c} \frac{\partial W_{c}}{\partial t}=\Sigma_{i=1}^{N}\Sigma_{\alpha =1}^{3}\left[\frac{\partial V}{\partial x_{i,\alpha }}\frac{\partial W_{c}}{\partial q_{i,\alpha }}-\frac{q_{i,\alpha }}{m}\frac{\partial W_{c}}{\partial x_{i,\alpha }}\right] \end{array}$$

Let [n] denote a set of non-negative integers (n(i=1,α=1),…,n(i=N,α=3)) and let $n=\Sigma_{l=1}^{N}\Sigma_{\alpha =1}^{3}n\left(l,\alpha \right)$. Let $\left[dq\right]=\prod_{i=1}^{N}\prod_{\alpha =1}^{3}dq_{i,\alpha }$. We introduce non-equilibrium moments $W_{c}\left(\left[n\right]\right)$ of *W* (using products of Hermite polynomials, by generalizing Equation (Equation 3)):(25)$$\begin{array}{c} \int \left[dq\right]\prod_{i=1}^{N}\prod_{\alpha =1}^{3}\frac{H_{n(i,\alpha )}\left(q_{i,\alpha }/{\left(2mk_{B}T\right)}^{1/2}\right)}{{\left(\pi^{1/2}2^{n(i,\alpha )}n\left(i,\alpha \right)!\right)}^{1/2}}W_{c}\left(\left(x\right),\left(q\right),t\right)\equiv W_{c}\left(\left(x\right);\left[n\right];t\right)=W_{c}\left(\left[n\right]\right)\;, \end{array}$$

If $W_{c}=W_{c,eq}$, then $W_{c,eq}\left(\left[0\right]\right)$ ([0]=(0,0,,,0)) is proportional to exp[−βV] and $W_{c,eq}\left(\left[n\right]\right)=0$, [n]≠[0] (say, n≠0). Equation (Equation 25) can also be applied to $W_{c,in}$ and gives the corresponding initial moments, $W_{c,in}\left(\left[n\right]\right)$. We shall work with the symmetrized moments $g\left(\left[n\right]\right)=W_{c,eq}{\left(\left[0\right]\right)}^{-1/2}W_{c}\left(\left[n\right]\right)$.

One gets an infinite reversible three-term linear recurrence for g([n])’s, generalizing Equations (Equation 5) and (Equation 6). It reads, as a result:(26)$$\begin{array}{c} \frac{\partial g(n(1,1),\ldots ,\;n(j,\beta ),\ldots ,\;n(N,3))}{\partial t}=-\Sigma_{l=1}^{N}\Sigma_{\alpha =1}^{3}[M_{l,\alpha ;n(l,\alpha );+}g\left(n\left(1,1\right),\ldots ,\;n\left(l,\alpha \right)+1,\ldots ,\;n\left(N,3\right)\right)\;\\ +M_{l,\alpha ;n(l,\alpha );-}g\left(n\left(1,1\right),\ldots ,\;n\left(l,\alpha \right)-1,\ldots ,\;n\left(N,3\right)\right)] \end{array}$$

(27)$$\begin{array}{c} M_{l,\alpha ;n(l,\alpha );+}={\left[\frac{\left(n\left(l,\alpha \right)+1\right)k_{B}T}{m}\right]}^{1/2}\left[\frac{\partial }{\partial x_{l,\alpha }}-\frac{1}{2k_{B}T}\frac{\partial V}{\partial x_{l,\alpha }}\right] \end{array}$$

(28)$$\begin{array}{c} M_{l,\alpha ;n(l,\alpha );-}={\left[\frac{n\left(l,\alpha \right)k_{B}T}{m}\right]}^{1/2}\left[\frac{\partial }{\partial x_{l,\alpha }}+\frac{1}{2k_{B}T}\frac{\partial V}{\partial x_{l,\alpha }}\right] \end{array}$$

The Laplace transform of the hierarchy Equation (Equation 26), which is the actual many-particle counterpart of Equation (Equation 7), can be formally solved in terms of linear operators D[[n];s], which generalize the previous D[n;s]. For details, see [25]. All D[[n];s] are square matrices, due to the indices i=1,…,N and α=1,…,3. In turn, each matrix element in those square matrices is an integral operator, arising from the partial differential operators $M_{l,\alpha ;n(l,\alpha );+}$ and $M_{l,\alpha ;n(l,\alpha );-}$, as *l*, α and n(l,α) vary. The D[[n];s] fulfill the following formal hierarchy (which generalizes Equation (Equation 8)):(29)$$\begin{array}{c} D\left[\left[n\right];s\right]={\left[s-M_{+,[n+1]}D\left[\left[n+1\right];s\right]M_{-,[n]}\right]}^{-1} \end{array}$$

The linear operators $M_{\pm ,[n]}$ are rectangular matrices, the elements of which are formed out of the partial differential operators $M_{l,\alpha ;n(l,\alpha );+}$ and $M_{l,\alpha ;n(l,\alpha );-}$. $M_{+,[n+1]}$ can be shown to be the adjoint of $-M_{-,[n]}$ [25]. By iterating Equation (Equation 29) indefinitely, one can express formally the linear operator D[[n];s] as an operator-continued fraction, which depends on all partial differential operators $M_{l,\alpha ;n(l,\alpha );+}$ and $M_{l,\alpha ;n(l,\alpha );-}$ and generalizes the operator-continued fraction for D[n;s].

By generalizing the iterative arguments in Section 2.3, it follows that D[[n];ϵ] for ϵ>0 is a Hermitian operator with non-negative eigenvalues for $n\geq n_{0}\geq 1$ [25] (in general, for either V≠0 or V=0), as a result. No approximation has been performed so far.

Simplification—A few remarks for the case V≡0 may be clarifying. In the Laplace transform of Equation (Equation 26), let us perform a Fourier transformation from configuration space ($x_{1}$, ..., $x_{N}$) to wavevector space ($k_{1}$, ..., $k_{N}$)≡[k]. Let $e\left(k;N\right)\equiv k_{B}T\sum_{j=1}^{N}{\left(2m\right)}^{-1}k_{j}^{2}$. Then, the Fourier transform $D_{1}\left[\left[k\right];\left[n\right];s\right]$ of D[[n];s] for Res>0 is an ordinary continued fraction, given in Equations (Equation 12)–(Equation 14) with e(k) replaced by e(k;N). Then, with such a replacement, the properties of $D_{1}\left(k;n;s\right)$ given in Section 2.4 also hold for $D_{1}\left[\left[k\right];\left[n\right];s\right]$. Notice that $D_{1}\left[\left[k\right];\left[n\right];s=0\right]$ diverges as $e{\left(k;N\right)}^{-1/2}$ if e(k;N)→0. On the other hand, and contrary to what happened for one particle in one spatial dimension (recall the comment after Equation (Equation 17)), $\int \left[dk\right]D_{1}\left[\left[k\right];\left[n\right];s=0\right]$ converges near e(k;N)=0. It is open whether actual counterpart of Equation (11) (containing D[[n];ϵ] acting on various $M_{l,\alpha ;n(l,\alpha );-}g\left(n\left(1,1\right),\ldots ,\;n\left(l,\alpha \right)-1,\ldots ,\;n\left(N,d\right)\right)$’s), when integrated over [k]’s to come back to (x)-space, would converge at small [k]’s as ϵ→0, when V≠0 and all $v_{ij}\left(|x_{i}-x_{j}|\right)$ are repulsive and tend quickly to zero for large distances. The approximate ansatz in Section 2.4 can be extended readily to $D[\left[n_{0}\right];\epsilon ]$.

### 3.2. Long-Term Approximation, Irreversibility, Liapunov Function and BBGKY Hierarchy

Approximations—Despite the very involved structure of Equation (Equation 26), we argue that a simple long-term approximation can be performed in it, for $v_{ij}\geq 0$ (and vanishing quickly at large distances) and very large *N* (eventually, in the thermodynamical limit), which generalizes directly the one in Section 2.5. This approximation consists in fixing s=ϵ>0 (ϵ being small) in the whole hierarchy of operators D[[n];s], for any $n\left(=\Sigma_{l=1}^{N}\Sigma_{\alpha =1}^{3}n\left(l,\alpha \right)\right)\geq n_{0}\left(>0\right)$ which, then, become Hermitian operators D[[n];ϵ] with no negative eigenvalues. It is crucial that *s*-dependences be kept in D[[n];s], for $n<n_{0}$. Notice that the non-vanishing factors $n{\left(l,\alpha \right)}^{1/2}$ and ${\left(\left(n\left(l,\alpha \right)+1\right)\right)}^{1/2}$ in $M_{l,\alpha ;n(l,\alpha );-}$ and $M_{l,\alpha ;n(l,\alpha );+}$, respectively, tend to reduce, as the n(l,α)’s increase, the relative importance of having fixed s=ϵ and the contribution of the latter in the D[[n];s=ϵ]’s with $n\geq n_{0}$. This is a genuine feature of the D[[n];s=ϵ]’s. See [25], where it was shown that by imposing $n_{0}>2$, the long-term approximation is still exactly consistent with all (five) hydrodynamical balance equations. For simplicity, we discard all the initial moments $W_{c,in}\left(\left[n\right]\right)$ for $n\geq n_{0}$. We regard $D[\left[n_{0}\right];\epsilon ]$ as a fixed (*s*-independent) operator, yielding all $g(\left[n_{0}\right])$ in terms of all $g(\left[n_{0}-1\right])$. Moreover, after the above long-term approximation, we shall continue with the same initial condition $W_{c,in}\left(\left[0\right]\right)$ at t=0: it may amount to another kind of approximation.

All that leads to a closed approximate hierarchy for g([n])’s (with initial moments $W_{c,in}\left(\left[n\right]\right)$), with $n<n_{0}$, which appears to yield an approximate irreversible evolution towards thermal equilibrium at *T*. g([n])’s and, then, $W_{c}\left(\left[n\right]\right)$ relax the quicker the larger *n*, provided that $n<n_{0}$. $W_{c}\left(\left[0\right]\right)$ would dominate the approach towards equilibrium for t→+∞. See [25]. All that appears to work for fixed $D[\left[n_{0}\right];\epsilon ]$. An arrow of time would follow approximately in the present case, as discussed below.

Simplification—As an extreme example, let $n_{0}=1$ (which is strictly consistent only with the hydrodynamical balance equations for mass): see [25] for $n_{0}=2,3$. By making the above long-term approximation and taking inverse Laplace transforms, one finds directly the irreversible Smoluchowski-like equation for the [n=0] moment, which generalizes (Equation 22) ([n=0] meaning n(1,1)=0,…,n(j,β)=0,…,n(N,3)=0):(30)$$\begin{array}{c} \frac{\partial g([n=0])}{\partial t}=\Sigma_{l=1}^{N}\Sigma_{\alpha =1}^{3}M_{l,\alpha ;n(l,\alpha )=0;+}\times \\ (\Sigma_{l^{\prime }=1}^{N}\Sigma_{\alpha^{\prime }=1}^{3}{\left[D\left[\left[n=1\right];\epsilon \right]\right]}_{l,\alpha ;l^{\prime },\alpha^{\prime }}M_{l^{\prime },\alpha^{\prime };n\left(l^{\prime },\alpha^{\prime }\right)=1;-})g\left(\left[n=0\right]\right) \end{array}$$as a result. The operator D[[n=1];ϵ] (Hermitian, with non-negative eigenvalues) has, as a square matrix, the matrix elements ${\left[D\left[\left[n=1\right];\epsilon \right]\right]}_{l,\alpha ;l^{\prime },\alpha^{\prime }}$. The initial condition is $W_{c,eq}{\left(\left[0\right]\right)}^{-1/2}W_{c,in}\left(\left[0\right]\right)$.

Discussion and consequences—Both the exact hierarchy for the g([n])’s and the closed approximate one for them after the long-term approximation are genuinely different from the non-equilibrium classical BBGKY hierarchy [7,8]. In fact, in the latter, in the equation for the distribution function for *n* particles, one leaves unintegrated their position vectors and momenta, while those for the remaining N−n particles are integrated over. Moreover, such an equation also depends on the distribution function for n+1 particles but not on that for n−1 ones, a feature which, beyond the approximate framework of the standard Boltzmann equation (for n=1) [2,7,8], does not seem to shed much light on the long-term approach to thermal equilibrium for larger *n*. By contrast, in the equation for g([n]) in the actual non-equilibrium hierarchy based upon $W_{c,eq}$, the contributions from g([n+1])’s are neatly different from (for large *n*, approximately smaller by a factor $≃n^{-1/2}$ than) those coming from g([n−1])’s, at least in the long-term approximation [25].

The structure of (Equation 30), with D[[n=1];ϵ] replaced by a constant is similar to that of the linear Smoluchowski equation in the standard Rouse model for polymer dynamics [38].

One leading purpose of this section is to introduce the *t*-dependent Liapunov function, for generic $n_{0}\geq 1$:(31)$$\begin{array}{c} L=\frac{1}{2}\sum_{n=0}^{n_{0}-1}\int \prod_{i=1}^{N}\prod_{\alpha =1}^{3}dx_{i,\alpha }g{\left(\left[n\right]\right)}^{2} \end{array}$$

The integration over any $x_{i,\alpha }$ is performed in $-\infty <x_{i,\alpha }<+\infty$. One has:(32)$$\begin{array}{c} \partial L/\partial t\leq 0 \end{array}$$

This follows readily for $n_{0}=1$ by using Equation (Equation 30) and the fact that the operator D[[n=1];ϵ] is Hermitian, with non-negative eigenvalues. The generalization for $n_{0}>1$, under the long-term approximation, proceeds by employing Sections 4 and 5 in [25]: that generalization is direct, although rather cumbersome (*N* particles in three dimensions and higher order moments being involved), and will be omitted. Then, the actual *L* is a nondecreasing function in the time evolution. Equations (Equation 31) and (Equation 32) constitute other results. At equilibrium:(33)$$\begin{array}{c} L=\frac{1}{2}\int \prod_{i=1}^{N}\prod_{\alpha =1}^{3}dx_{i,\alpha }q_{eq}^{3N}\exp\left[-\beta V\right]\equiv L_{eq} \end{array}$$which is proportional to the standard exponential of the equilibrium entropy. There has been much interesting work aimed at defining, in different frameworks (thermodynamical, statistical...), a *t*-dependent entropy S(t) in non-equilibrium phenomena, in such a way that the inequality dS(t)/dt≥0 would hold. The aim would be to characterize in a general setting, the notion of evolution. For updated accounts, see [20]. It appears that at present, a different number of non-equilibrium entropies have been proposed (and do the job) in various frameworks separately [39]. However, to the best of the present author’s knowledge, no general agreement has been reached on a unique non-equilibrium entropy S(t) valid for different frameworks, and an accepted definition of it based on statistical mechanics appears also to be lacking, so far. Both in case that a general definition of non-equilibrium entropy does not exist or in case that it could indeed be formulated at the end (although it be unknown at present), a partial way out towards the characterization of an arrow of time is the following. Once the long-term approximation has been carried out as indicated above, we have obtained a *t*-dependent function *L* through Equation (Equation 31), the variation of which seems to be adequate to define an arrow of time, by virtue of Equation (Equation 32).

## 4. Quantum Particles: One Particle 

In recent times, various rewarding results have been obtained for a system subject to a larger “heat bath” in quantum mechanics, under various conditions. Some of those results (which are related to and support our present work) are: (i) derivation of the canonical density operator for the system for long time [40], and for a suitable overwhelming majority of wave functions (typicality) [41]; (ii) the system will approach an equilibrium state and remain close to it for almost all times, independently both on the hb and, under suitable conditions, also on the initial state of the system [42]. For equilibration of an isolated system, see [43]. Those results required to consider the quantum states of the system and of the hb altogether and rely on their reciprocal entanglement. For further works strengthening those results, see [44,45,46,47] and references therein. For general aspects on quantum open systems, see also [2,8,9,13,14,15,17,18,19] and references therein.

General assumption and simplification—Throughout our quantum study, we assume that the state of the system, for sufficiently long times, approaches towards its own canonical density operator at thermal equilibrium with the hb at absolute temperature *T*, as studied in [40,41,42,43,44,45,46,47]. To simplify matters, we will focus exclusively on the quantum states of the system considered.

Our study here of the quantum one particle case subject to an external potential will be addressed towards displaying analogies and genuine differences with respect to the preceding classical case (Section 2, mostly) and to provide some hints for the quantum *N* particle case (Section 5), and for those with N=2,3 (Section 6 and Section 7).

### 4.1. General Aspects

We shall consider one non-relativistic quantum spinless Brownian particle of mass m(>0) and momentum operator −iℏ(∂/∂x), in one spatial dimension *x*, with (Hermitian) quantum Hamiltonian (*ℏ* being Planck’s constant):(34)$$\begin{array}{c} H=-\frac{\hbar^{2}}{2m}\frac{\partial^{2}}{\partial x^{2}}+V \end{array}$$

Assumptions—The real potential V(x) fulfills: V(x)→0 quickly, as ∣x∣→+∞ and V(x) and all $d^{n}V\left(x\right)/dx^{n}$, n=1,2,3,…, are continuous for any *x*. Also, having in mind possible generalizations, we suppose that V(x)=V(−x) and allow for V(x)<0 and, hence, the possibility of bound states of the particle by the potential. The particle is also separately subject to the hb.

The non-equilibrium statistical evolution for t>0 is given by the density operator ρ=ρ(t) (a statistical mixture of quantum states), with the initial condition $\rho \left(t=0\right)=\rho_{in}$. ρ(t) for t>0 and $\rho_{in}$ are Hermitian and positive-definite linear operators acting in the Hilbert space spanned by the set of all eigenfunctions $\varphi_{j}\left(x\right)$ of *H*. Unless otherwise stated, we shall not impose that ρ(t) be normalized. One has ([H,ρ]=Hρ−ρH):(35)$$\begin{array}{c} \frac{\partial \rho }{\partial t}=\frac{1}{i\hbar }\left[H,\rho \right] \end{array}$$

We consider the matrix element 〈x−y|ρ(t)|x+y〉 of ρ(t) in generic eigenstates, |x−y〉, |x+y〉, of the quantum position operator. The quantum Wigner function W=W(x,q;t), determined by ρ, is [48,49,50,51,52]:(36)$$\begin{array}{c} W\left(x,q;t\right)=\frac{1}{\pi \hbar }\int_{-\infty }^{+\infty }dy\exp\left[\frac{i2qy}{\hbar }\right]\langle x-y|\rho \left(t\right)|x+y\rangle \end{array}$$

The initial non-equilibrium Wigner function at t=0 is $W_{in}$, given by Equation (Equation 36) if $\rho =\rho_{in}$. For t>0, the exact dissipationless quantum master equation for *W* [48,49] is: (37)$$\begin{array}{c} \frac{\partial W(x,q;t)}{\partial t}=-\frac{q}{m}\frac{\partial W(x,q;t)}{\partial x}+M_{Q}W \end{array}$$
(38)$$\begin{array}{c} M_{Q}W=\int_{-\infty }^{+\infty }dq^{\prime }W\left(x,q^{\prime };t\right)\int_{-\infty }^{+\infty }\frac{idy}{\pi \hbar^{2}}\left[V\left(x+y\right)-V\left(x-y\right)\right]\\ \times \exp\left[\frac{i2(q-q^{\prime })y}{\hbar }\right]=\frac{dV}{dx}\frac{\partial W}{\partial q}-\frac{\hbar^{2}}{3!2^{2}}\frac{d^{3}V}{dx^{3}}\frac{\partial^{3}W}{\partial q^{3}}+\ldots \end{array}$$

As ℏ→0, Equation (Equation 37) becomes formally, by dropping all *ℏ*-dependent terms (containing $\partial^{n}W/\partial q^{n}$, n=3,5,…) in Equation (Equation 38), the classical Liouville Equation (Equation 1), with $W\to W_{c}$ [48,49]. All terms in the series in Equation (Equation 38) contribute, in general, for the assumed *V*’s.

Assumption. As ∣q∣→+∞, W(x,q;t)→0 quickly, for fixed *x* and *t*, so that $\int_{-\infty }^{+\infty }dqW\left(x,q;t\right)q^{n}$ converges, for any integer n≥0 [32,34].

Under the latter assumption, Equation (Equation 37) readily implies that $\left(\partial /\partial t\right)\int_{-\infty }^{+\infty }dx\int_{-\infty }^{+\infty }dqW\left(x,q;t\right)=0$.

See [19] for theorems and constructive methods to find stationary solutions of Equation (Equation 35).

We shall now consider the equilibrium Wigner function $W_{eq}$, fulfilling Equations (Equation 39) and accounting for thermal equilibrium at *T* with the hb. Like in the classical case, the solutions of Equations (Equation 37) and (38) are not expected to approach $W_{eq}$ exactly, unless some approximation be made. $W_{eq}$ arises from the canonical (*t*-independent) density operator $\rho_{eq}=\exp\left[-\beta H\right]$, through Equation (Equation 36). $W_{eq}\left(x,q\right)$ is neither Gaussian in *q* nor known in closed form in general [48,49] and their dependences on *q* and *x* do not factorize. One has:(39)$$\begin{array}{c} -\frac{q}{m}\frac{\partial W_{eq}}{\partial x}+M_{Q}W_{eq}=0,\;\frac{\partial W_{eq}}{\partial t}=0 \end{array}$$

### 4.2. Novel Features: $W_{eq}$ as Quasi-Definite Functional in Momentum and Orthogonal Polynomials

We now remind the known quantum difficulty: neither *W* nor $W_{eq}$ can be warranted to be ≥0 (negativity), in general [48,49]. A necessary and sufficient condition for the Wigner function associated with a Schrodinger wave function be ≥0 is that the latter be the exponential of a quadratic polynomial [53]. The domain in which $W_{eq}<0$ may occur has be consistent with $\int_{-\infty }^{+\infty }dxW_{eq}\geq 0$ and $\int_{-\infty }^{+\infty }dqW_{eq}\geq 0$.

Despite W<0, we shall now invoke, in outline, an acceptable mathematical framework based upon the theory of orthogonal polynomials [54]. Let us consider a kernel K=K(y) (which could be ≤0), a set of functions f=f(y) and the following functional $L_{K}$ determined by the kernel *K*: $f\to L_{K}\left[f\right]=\int_{-\infty }^{+\infty }dyK\left(y\right)f\left(y\right)$. We shall suppose that all integrals over *y* are convergent. Let us consider, successively: $\mu_{n}=L_{K}\left[y^{n}\right]$, n=0,1,2,3,…, the set of all (S+1)×(S+1) matrices $M_{S}$ (S=0,1,2,3,…) with (i,j)-th element equal to $\mu_{i+j}$ (i,j=0,…,S), and their determinants: $Det[M_{S}]$. By definition, the functional $L_{K}$ is quasi-definite if $Det[M_{S}]\neq 0$ for any S=0,1,2,3,… [54]. If $L_{K}$ is a quasi-definite functional, then a theorem [54] implies the existence of a family of orthogonal polynomials, named here and below $H_{Q,n}=H_{Q,n}\left(y\right)$, with weight function *K* (even if K<0 in some domain in *y*).

Assumption—Let $y=q/q_{eq}$ ($q_{eq}={\left(2m/\beta \right)}^{1/2}$). Recall that we have assumed that *V* and all its derivatives exist and are continuous. Then (by extending directly [32]), one can accept that regarding their *y*-dependences, the Wigner function *W* and $W_{eq}$ determine, respectively, quasi-definite functionals $L_{W}$ (for any *x* and *t*) and $L_{W_{eq}}$ (for any *x*). We shall suppose the validity of the latter property in all that follows. Its interest is obvious: if it holds (as we suppose), it implies the existence of orthogonal polynomials. To prove that assumption for any *W* and $W_{eq}$, in general, lies outside our scope: its validity for $L_{W_{eq}}$ has been checked (and confirmed) in one interesting case in [32].

We shall introduce the (unnormalized) polynomials in *y* ($=q/q_{eq}$), $H_{Q,n}=H_{Q,n}\left(y\right)$ (n=0,1,2,3,…), orthogonalized in *y* (for fixed *x*) by using $W_{eq}$ as weight function. By choosing $H_{Q,0}\left(q\right)=1$, for $n\neq n^{\prime }$ and any *x* (left unintegrated), we impose:(40)$$\begin{array}{c} \int_{-\infty }^{+\infty }dyW_{eq}H_{Q,n}\left(y\right)H_{Q,n^{\prime }}\left(y\right)=0 \end{array}$$

The $H_{Q,n}$’s, depending parametrically on *x* for n≥1, will be used for the time evolution [32]. The normalized polynomials are $H_{Q,n}/h_{Q,n}^{1/2}$ ($h_{Q,n}=\int_{-\infty }^{+\infty }dyW_{eq}H_{Q,n}{\left(y\right)}^{2}$). We shall look for the $H_{Q,n}\left(y\right)$’s as:(41)$$\begin{array}{c} H_{Q,n}\left(y\right)=y^{n}+\sum_{j=1}^{n}\epsilon_{Q,n,n-j}y^{n-j} \end{array}$$$\epsilon_{Q,n,n-j}$, *y*-independent (but *x*-dependent, in general). One has $\epsilon_{Q,n,n-j}=0$ for odd *j*, so that $H_{Q,n}\left(-y\right)={\left(-1\right)}^{n}H_{Q,n}\left(y\right)$. The simplest $\epsilon_{Q,n,n-j}=0$ is: $\epsilon_{Q,2,0}=-\left[\int_{-\infty }^{+\infty }dyW_{eq}y^{2}\right]/\left[\int_{-\infty }^{+\infty }dyW_{eq}\right]$. We omit further details of $\epsilon_{Q,n,n-j}$: see Subsection 4.4 and Appendix A in [32].

### 4.3. Non-Equilibrium Moments and Hierarchy 

We shall analyze general off-equilibrium situations by using Equations (Equation 37) and (Equation 38). Using for convenience the unnormalized $H_{Q,n}\left(y\right)$’s in Equation (Equation 41) (n=0,1,2,…), we introduce the new non-equilibrium moments:(42)$$\begin{array}{c} W_{n}=W_{n}\left(x;t\right)=\int_{-\infty }^{+\infty }dyH_{Q,n}\left(y\right)W \end{array}$$

The initial condition $W_{in,n}$ for $W_{n}$ is obtained by replacing *W* by $W_{in}$ in Equation (Equation 42). One has the following (formal) expansion for *W*, which generalizes Equation (Equation 4):(43)$$\begin{array}{c} W=W_{eq}\left(x,q\right)\sum_{n=0}^{+\infty }W_{n}\left(x;t\right)\frac{H_{Q,n}\left(y\right)}{h_{Q,n}} \end{array}$$

For $W=W_{eq}\left(x,q\right)$, Equation (Equation 42) yields $W_{eq,n}=0$ if n>0, and $W_{eq,0}=h_{Q,0}$. The transformation of Equations (Equation 37) and (Equation 38) into a linear hierarchy for the new moments $W_{n}$ can be carried out as in the classical case, but it requires far more work. The structure of the general non-equilibrium hierarchy is: (44)$$\begin{array}{c} \frac{\partial W_{n}}{\partial t}=-M_{n,n+1}W_{n+1}-\sum_{n^{\prime }=1}^{n}M_{n,n-n^{\prime }}W_{n^{\prime }} \end{array}$$(45)$$\begin{array}{c} M_{n,n+1}W_{n+1}\equiv \frac{q_{eq}}{m}\frac{\partial W_{n+1}}{\partial x} \end{array}$$(46)$$\begin{array}{c} M_{n,n-1}W_{n-1}=\frac{q_{eq}}{m}\left[\left(\epsilon_{Q,n+1,n-1}-\epsilon_{Q,n,n-2}\right)\frac{\partial W_{n-1}}{\partial x}+\left(\frac{\partial \epsilon_{Q,n,n-2}}{\partial x}\right)W_{n-1}\right]+\frac{n}{q_{eq}}\frac{\partial V}{\partial x}W_{n-1}. \end{array}$$$M_{n,n^{\prime }=0}=0$ for any *n*, except for n=1 (with $n^{\prime }=0$). In [32,34], we have obtained the first five equations in that quantum hierarchy. For any V≠0, the equation for $\partial W_{n}/\partial t$ for any *n* contains a contribution from $W_{n+1}$ (always with the same structure, with *n*-independent coefficients) given in Equation (Equation 45), but does not on higher order moments $W_{n^{\prime }}$, with $n^{\prime }>n+1$. On the other hand, the contribution from $W_{n^{\prime }}$ ($0<n^{\prime }\leq n-1$) does carry *n*-dependent coefficients, which increase with *n*. The quantum hierarchy is not a three-term one, due to moments $W_{n^{\prime }}$ with $n^{\prime }<n$: the lowest order at which this can be displayed occurs for n=4, where the full quantum equation does contain a term of order $\hbar^{3}$ in $W_{1}$. Equations (Equation 44)–(Equation 46) constitute another result. No approximation has been carried out, so far, in this section for brevity, as they will be performed in the following sections. See [32,34] for details and further analysis and approximations in the one-dimensional case (in particular, for the long term).

## 5. N(>1) Quantum Particles: Equilibrium and Non-Equilibrium Statistical Distributions

### 5.1. General Aspects and Factoring Out the Center of Mass

We shall now consider *N* non-relativistic spinless quantum particles of masses $m_{j}$ (with j=1,…,N) in three spatial dimensions, inside a large finite volume Ω. Those individual particles can be either atoms or small molecules. In cartesian coordinates, the position vector and the momentum operator of the *j*-th particle are $x_{j}=\left(x_{j,1},x_{j,2},x_{j,3}\right)$ and $p_{j}=\left(p_{j,1},p_{j,2},p_{j,3}\right)$, where $p_{j,\alpha }=-i\hbar \left(\partial /\partial x_{j,\alpha }\right)$, α=1,2,3. We shall denote $\left(x\right)=\left(x_{1},\ldots ,\;x_{N}\right)$.

Assumptions—We emphasize that Ω (outside which, an infinitely repulsive potential is assumed) is large only at the microscopic scale. We assume that there are no external potentials and that all interactions acting on the particles are described, in principle, by time-independent and velocity-independent instantaneous potentials *V* among the particles. V=V((x)) is the sum of two-body potentials $v_{ij}=v_{ij}\left(\right|x_{j}-x_{i}\left|\right)$, as in Section 3. We assume that $v_{ij}\left(r\right)$ and all $d^{n}v_{ij}\left(r\right)/dr^{n}$, for n=1,2,3,…, are continuous for all r>0: compare with Section 4. The total (Hermitian) quantum Hamiltonian is

(47)$$\begin{array}{c} H=-\sum_{j=1}^{N}\sum_{\alpha =1}^{3}\frac{\hbar^{2}}{2m_{j}}\frac{\partial^{2}}{\partial x_{j,\alpha }^{2}}+V\left(\left(x\right)\right). \end{array}$$

Let the *N*-particle system be immersed in a hb at thermal equilibrium, at absolute temperature *T*.

We resort to the non-equilibrium Wigner function [2,8,48,49,50,52] for the *N* particles, which reads (in terms of the *N*-particle density operator ρ(t)):(48)$$\begin{array}{c} W\left(\left(x\right),\left(q\right);t\right)=\frac{1}{{\left(\pi \hbar \right)}^{3N}}\int \left({\mathrm{dx}}^{\prime }\right)\exp\left[\frac{2i}{\hbar }\sum_{j=1}^{N}\sum_{\alpha =1}^{3}x_{i,\alpha }^{\prime }q_{i,\alpha }\right]\langle \left(x\right)-\left(x^{\prime }\right)|\rho \left(t\right)|\left(x\right)+\left(x^{\prime }\right)\rangle , \end{array}$$where $\left(q\right)=\left(q_{1},\ldots ,\;q_{N}\right)$ and $\left({\mathrm{dx}}^{\prime }\right)=\prod_{i=1}^{N}\prod_{\alpha =1}^{3}dx_{i,\alpha }^{\prime }$.

The time evolution of the Wigner function is furnished by
(49)$$\begin{array}{c} \frac{\partial W((x),(q);t)}{\partial t}=-\sum_{j=1}^{N}\sum_{\alpha =1}^{3}\frac{q_{j,\alpha }}{m_{j}}\frac{\partial W((x),(q);t)}{\partial x_{j,\alpha }}+M_{Q}W, \end{array}$$with
(50)$$\begin{array}{cc} M_{Q}W & =\frac{i}{\hbar {\left(\pi \hbar \right)}^{3N}}\int \left({\mathrm{dq}}^{\prime }\right)W\left(\left(x\right),\left(q^{\prime }\right);t\right)\int \left({\mathrm{dx}}^{\prime }\right)\exp\;\left[\frac{2i}{\hbar }\sum_{j=1}^{N}\sum_{\alpha =1}^{3}x_{j,\alpha }^{\prime }\left(q_{j,\alpha }-q_{j,\alpha }^{\prime }\right)\right]\\ & \times \left[V\left(\left(x\right)+\left(x^{\prime }\right)\right)-V\left(\left(x\right)-\left(x^{\prime }\right)\right)\right], \end{array}$$where $\left({\mathrm{dq}}^{\prime }\right)=\prod_{i=1}^{N}\prod_{\alpha =1}^{3}dq_{i,\alpha }^{\prime }$.

Approximation—As Ω is large, we shall approximate spatial integrals by those for an infinite volume when such approximations are harmless, unless some specific discussion be required. Strictly speaking, as Ω is not infinite, the $\left(q^{\prime }\right)$ are discretized momenta and $\int ({\mathrm{dq}}^{\prime })$ in Equation (Equation 50) should be interpreted as a 3N-fold series. However, as Ω is large, we shall disregard the small spacings in that 3N-fold series and approximate it as a 3N-fold integral. We shall accept that all integrals (or all series) over momenta converge for large values of the latter: explicit computations do support it.

We shall suppose a quantum initial condition, determined by some density matrix at t=0, $\rho_{in}$, eventually suitable for chemical reactions. The initial Wigner function $W_{in}\left(\left(x\right),\left(q\right)\right)$ is determined by $\rho_{in}$ through Equation (Equation 48). The canonical density operator describing the *N*-particle system at thermal equilibrium with the hb is $\rho_{eq}=\exp\left[-\beta H\right]$ (with $\partial \rho_{eq}/\partial t=0$). The equilibrium Wigner function $W_{eq}\left(\left(x\right),\left(q\right)\right)$ is given by Equation (Equation 48) for the actual equilibrium density operator $\rho_{eq}$.

Let $P=\sum_{j=1}^{N}p_{j}$ be the total momentum. Then, one has: [H,P]=0 ([A,B]=AB−BA being the commutator of the operators *A* and *B*).

For convenience, the center of mass (CM) degrees of freedom will be factored out from the relative ones from the outset. The standard CM position vector is $X=M^{-1}\left[\sum_{j=1}^{N}m_{j}x_{j}\right]$ with $M=\sum_{j=1}^{N}m_{j}$. Then, one has

(51)$$\begin{array}{c} H=H_{CM}+H_{N,re},\;H_{CM}=-\frac{\hbar^{2}}{2M}\frac{\partial^{2}}{\partial X^{2}},\;\left[H_{CM},H_{N,re}\right]=0, \end{array}$$

Here, $H_{CM}$ is the free CM Hamiltonian and $H_{N,re}$ is the Hamiltonian for the remaining (internal) relative position vectors: see Section 6 and Section 7 for N=2 and N=3, respectively. $\rho_{eq}$ factorizes as $\rho_{eq}=\rho_{CM,eq}⊗\rho_{N,re,eq}$, where $\rho_{CM,eq}=\exp\left[-\beta H_{CM}\right]$ and $\rho_{N,re,eq}=\exp\left[-\beta H_{N,re}\right]$. The actual $W_{eq}=W_{eq}\left(\left(x\right),\left(q\right)\right)$ factorizes as
(52)$$\begin{array}{c} W_{eq}\left(\left(x\right),\left(q\right)\right)=W_{CM,eq}W_{N,re,eq}, \end{array}$$$W_{N,re,eq}$ being the relative equilibrium Wigner function. Spatial integrations over ***X*** is carried out inside the volume Ω. $W_{CM,eq}$ is proportional to $\exp\left[-\beta (Q^{2}/2M)\right]$ ($Q=\sum_{j=1}^{N}q_{j}$). Correspondingly, by using the assumptions below on the initial conditions, there is also factorization off-equilibrium: $W\left(\left(x\right),\left(q\right);t\right)=W_{CM}\left(X,Q;t\right)W_{N,re}$, with $\partial W_{CM}/\partial t=-\left(Q/M\right)\partial W_{CM}/\partial X$, and the (*t*-reversible) Wigner function $W_{N,re}$ of the relative degrees of freedom. Both $W_{N,re,eq}$ and $W_{N,re}$ depend on N−1 relative position vectors and momenta, denoted collectively as ${\left(x\right)}_{N-1,re}$ and ${\left(q\right)}_{N-1,re}$. In general, for given *N*, there are several interesting choices for the sets ${\left(x\right)}_{N-1,re}$ and ${\left(q\right)}_{N-1,re}$ out of the (x)’s and (q)’s, respectively: see, in particular, Section 7.

Assumptions—It is a reasonable assumption that the initial conditions $\rho_{in}$ and $W_{in}\left(\left(x\right),\left(q\right)\right)$ factorize into CM and relative ones. The initial condition for $W_{N,re}$ is $W_{N,re,in}$. Neither $W_{eq}$ nor $W_{N,re,eq}$ can be warranted to be non-negative in general [48,49,52]. $W_{\mathrm{CM},\mathrm{eq}}$, Gaussian in momenta, determines an infinite family of orthogonal polynomials (the standard Hermite ones) in momenta. We shall assume that regarding its ${\left(q\right)}_{N-1,re}$-dependence, $W_{N,re,eq}$ determines quasi-definite functionals (for any ${\left(x\right)}_{N-1,re}$), by extending Section 4. This assumption implies the existence of an infinite family of orthogonal polynomials generated by $W_{N,re,eq}$.

### 5.2. Orthogonal Polynomials, Non-Equilibrium Moments and Hierarchy

We shall consider one choice ${\left(q\right)}_{N-1,re}$, and let $q_{N-1,re,j}$, j=1,…,N−1 be the momenta constituting the set ${\left(q\right)}_{N-1,re}$. Also, let $m_{re,j}$ be some suitable masses, associated to the choice ${\left(q\right)}_{N-1,re}$ and defined out of all $m_{1},\ldots ,m_{N}$. See Section 6 and Section 7. Let $q_{eq,j}={\left(2m_{re,j}/\beta \right)}^{1/2}$ (independent of (x)) and let ${\left(y\right)}_{N-1,re}$ be the set formed by all $q_{eq,j}^{-1}q_{N-1,re,j}$, j=1,…,N−1.

We consider the equilibrium distribution $W_{N,re,eq}\left({\left(x\right)}_{N-1,re},{\left(q\right)}_{N-1,re}\right)$ and the orthogonal polynomials $H_{\left([n]\right)}\left({\left(y\right)}_{N-1,re}\right))$ (depending parametrically on ${\left(x\right)}_{N-1,re}$), generated by the former as weight function. Here, ([n]) denotes N−1 (in general, different) triplets of non-negative integers $\left[n\right]=\left(n_{1},n_{2},n_{3}\right)$: the *j*-th triplet [n] is associated with the *j*-th momentum in the set ${\left(q\right)}_{N-1,\mathrm{re}}$. Use will also be made of additional obvious short-hand notations: ([0])=([0],[0],….), ${\left(\left[1\right]\right)}_{j,\alpha }$ with 1 in the α-th (=1,2,3) position of the *j*-th triplet with all other components of the remaining triplets being zero and ${\left(\left[2\right]\right)}_{j,\alpha }$ similar to ${\left(\left[1\right]\right)}_{j,\alpha }$ but with a 2 in the α-th position of the *j*-th triplet.

The orthogonal polynomials $H_{\left([n]\right)}$ are chosen with $H_{\left([0]\right)}\left({\left(y\right)}_{N-1,re}\right)=1$ and: (53)$$\begin{array}{c} \int {\left(dy\right)}_{N-1,re}W_{N,re,eq}\left({\left(x\right)}_{N-1,re},{\left(q\right)}_{N-1,re}\right)H_{\left([n]\right)}\left({\left(y\right)}_{N-1,re}\right)H_{\left(\left[n^{\prime }\right]\right)}\left({\left(y\right)}_{N-1,re}\right)=0, \end{array}$$${\left(\mathrm{dy}\right)}_{N-1,re}=\prod_{i=1}^{N-1}\prod_{\alpha =1}^{3}dy_{i,\alpha }$, $y_{i,\alpha }$ being the α-th (=1,2,3) component of the *i*-th $q_{eq,j}^{-1}q_{N-1,re,i}$. $H_{{\left(\left[1\right]\right)}_{j,\alpha }}=H_{{\left(\left[1\right]\right)}_{j,\alpha }}\left(y_{j,\alpha }\right)=y_{j,\alpha }$ and $H_{{\left(\left[2\right]\right)}_{j,\alpha ,l,\gamma }}=y_{j,\alpha }y_{l,\gamma }+\epsilon_{{\left(\left[2\right]\right)}_{j,\alpha ,l,\gamma }}$, $\epsilon_{{\left(\left[2\right]\right)}_{j,\alpha ,l,\gamma }}$ being ${\left(y\right)}_{N-1,re}$ but dependent on ${\left(x\right)}_{N-1,re}$: see Appendix A and Appendix B. The non-equilibrium moments of the actual non-equilibrium relative Wigner function read: (54)$$\begin{array}{c} W_{\left([n]\right)}\equiv W_{\left([n]\right)}\left({\left(x\right)}_{N-1,\mathrm{re}};t\right)=\int {\left(dy\right)}_{N-1,re}H_{\left([n]\right)}\left({\left(y\right)}_{N-1,re}\right)W_{N,re}\left({\left(x\right)}_{N-1,re},{\left(q\right)}_{N-1,re};t\right). \end{array}$$and so on for the initial non-equilibrium condition for the actual relative degrees of freedom, to be denoted as $W_{in,([n])}$ for simplicity. The transformation of Equations (Equation 49) and (Equation 50) (after having factored out the CM) and Equation (Equation 54) into an infinite linear hierarchy for the non-equilibrium moments $W_{\left([n]\right)}$ can be carried out as in previous cases. The structure of the general (*t*-reversible) linear non-equilibrium hierarchy, a genuine consequence of quantum mechanics, reads:(55)$$\begin{array}{c} \frac{\partial W_{\left([n]\right)}}{\partial t}=-\sum_{j=1}^{N-1}\sum_{\gamma =1}^{3}M_{\left(\left[n\right]\right),{\left(\left[n+1\right]\right)}_{j,\gamma }}W_{{\left(\left[n+1\right]\right)}_{j,\gamma }}-\sum_{\left(\left[n^{\prime }\right]\right)}M_{\left(\left[n\right]\right),\left(\left[n-n^{\prime }\right]\right)}W_{\left(\left[n-n^{\prime }\right]\right)}, \end{array}$$with
(56)$$\begin{array}{c} M_{\left(\left[n\right]\right),{\left(\left[n+1\right]\right)}_{j,\gamma }}W_{{\left(\left[n+1\right]\right)}_{j,\gamma }}\equiv \frac{q_{eq,j}}{m_{re,j}}\frac{\partial W_{{\left(\left[n+1\right]\right)}_{j,\gamma }}}{\partial x_{j,\gamma }}. \end{array}$$

Equation (Equation 55) is a result which generalizes Equation (Equation 44). The subscript ${\left(\left[n+1\right]\right)}_{j,\gamma }$ in $W_{{\left(\left[n+1\right]\right)}_{j,\gamma }}$ denotes that one unity has been added to the integer in the (j,γ) location, and analogously for the corresponding interpretation of the sum $\sum_{\left(\left[n^{\prime }\right]\right)}$ in Equation (Equation 44): notice that the non-negative integers in $\left(\left[n^{\prime }\right]\right)$ grow until $\sum_{j=1}^{N-1}\sum_{\alpha =1}^{3}n_{j\alpha }^{\prime }=\sum_{j=1}^{N-1}\sum_{\alpha =1}^{3}n_{j\alpha }$. Further explanations are omitted, see [34]. Use will be made of:(57)$$\begin{array}{c} M_{{\left(\left[1\right]\right)}_{j,\alpha },\left(\left[0\right]\right)}W_{\left([0]\right)}=-\frac{q_{eq,j}}{m_{re,j}}\sum_{l=1}^{N-1}\sum_{\gamma =1}^{3}\frac{\partial }{\partial x_{l,\gamma }}\left(\epsilon_{{\left(\left[2\right]\right)}_{j,\alpha ;l,\gamma }}W_{\left([0]\right)}\right)+\frac{1}{q_{eq,j}}\frac{\partial V}{\partial x_{j,\alpha }}W_{\left([0]\right)}. \end{array}$$(58)$$\begin{array}{c} \epsilon_{{\left[2\right]}_{j,\alpha ,l,\gamma }}=-\frac{\int {\left(dy\right)}_{N-1,re}W_{N,re,eq}\left({\left(x\right)}_{N-1,re},{\left(q\right)}_{N-1,re}\right)y_{j,\alpha }y_{l,\gamma }}{\int {\left(dy\right)}_{N-1,re}W_{N,re,eq}\left({\left(x\right)}_{N-1,re},{\left(q\right)}_{N-1,re}\right)}, \end{array}$$

Equations (A1) and (A25) will be particular cases of Equation (58). The quantum Equation (Equation 55) is also a counterpart of the classical Equation (Equation 26). We emphasize three crucial difference between them. First, in the classical Equation (Equation 26), the factoring out of $W_{c,eq,0}{\left(x\right)}^{-1/2}$ enabled to work with the symmetrized moments $g_{n}$, while no similar factorization is possible in Equation (Equation 55), so that one has to deal with the $W_{\left([n]\right)}$’s. Second, the CM has not been factored out in the classical Equation (Equation 26), while it has indeed been in the quantum Equation (Equation 55). Third, in Equation (Equation 26) the orthogonal polynomials were independent of (x) and, consequently, the only dependences on the two-body potentials are those displayed explicitly in the right-hand-side of Equation (Equation 26). In the quantum Equation (Equation 55), the orthogonal polynomials are ${\left(x\right)}_{N-1,\mathrm{re}}$-dependent, and so are $\epsilon_{{\left(\left[2\right]\right)}_{j,\alpha },l,\gamma }$ and, in general, the coefficients of $H_{\left([n]\right)}$ contained in $M_{\left(\left[n\right]\right),\left(\left[n-n^{\prime }\right]\right)}$. All coefficients in (Equation 55) are expressed in terms of *V* and of quantities computed out of the equilibrium solution $W_{N,re,eq}$. When $W_{N,re}=W_{N,re,eq}$, all equilibrium moments $W_{eq,([n])}$ with ([n])≠([0]) vanish, except $W_{eq,([0])}$. Moreover, $W_{eq,([0])}\left(\neq 0\right)$ and $W_{eq,([n])}=0$ for ([n])≠([0]) solve the hierarchy Equation (Equation 55), with all $\partial W_{eq,([n])}/\partial t=0$.

### 5.3. Approximations: Small Thermal Wavelength and Long-Term

In the *t*-evolution, not far from thermal equilibrium, and recalling [40,41,42,43,44,45,46,47], one expects that $W_{\left([0]\right)}$ would be dominant and that any $W_{\left([n]\right)}$, for any ([n])≠([0]), be small (the more negligible the larger *t* and $\sum_{j=1}^{N-1}\sum_{\alpha =1}^{3}n_{j\alpha }$ are).

Assumption and approximation—We consider that *T* is not high (so as not to enter fully in the classical regime). We shall also avoid the very low temperature regime. Therefore, we shall treat the regime of typical chemical reactions: small thermal wavelength. Then, the general non-equilibrium hierarchy Equation (Equation 55) can be approximated by a three-term hierarchy, namely:(59)$$\begin{array}{c} \frac{\partial W_{\left([n]\right)}}{\partial t}=-\sum_{j=1}^{N-1}\sum_{\gamma =1}^{3}M_{\left(\left[n\right]\right),{\left(\left[n+1\right]\right)}_{j,\gamma }}W_{{\left(\left[n+1\right]\right)}_{j,\gamma }}-\sum_{}M_{\left([n]),([n-1]\right)}W_{\left([n-1]\right)}, \end{array}$$

The short-hand notation $\sum_{}$ in Equation (Equation 59) means that only $\left[n^{\prime }\right]=\left[1\right]$ contributes (compared to $\sum_{\left(\left[n^{\prime }\right]\right)}$ in Equation (Equation 55), where various $\left[n^{\prime }\right]$ have to be considered). That short-hand notation omits, for brevity, dependences on certain indices: compare with the more explicit Equation (Equation 67). The justification of the above small thermal wavelength, leading from Equations (Equation 55)–(Equation 59) will be omitted, as it is a straightforward, but cumbersome, generalization of the one to be given in Section 6.2 for N=2. We emphasize that Equation (Equation 59) is still a quantum hierarchy, since it contains the quantum-mechanical $M_{\left(\left[n\right]\right),{\left(\left[n-1\right]\right)}_{j,\gamma }}$.

The solution of the resulting approximate non-equilibrium three-term hierarchy Equation (Equation 59) is given, through a Laplace transform from *t* to the variable *s*, in terms of products of certain *s*-dependent generalized operator-continued fractions D[([n]);s], which constitute extensions of Equations (Equation 8) and (Equation 29).

Assumption and approximation—We proceed to the long-term approximation (thus introducing irreversibility) for *t* longer than a certain largest effective evolution time. We shall omit details, which constitute a formal extension of those in Section 2.5. We shall assume the initial condition $W_{in,([0])}\neq 0$ and $W_{in,([n])}=0$ for ([n])≠([0]).

Upon taking the inverse Laplace transform, one gets (for fixed and small s=ϵ>0) the following irreversible approximate quantum (Smoluchowski-like) equation for the lowest non-equilibrium moment $W_{\left([0]\right)}\left(x;t\right)$
(60)$$\begin{array}{c} \frac{\partial W_{\left([0]\right)}}{\partial t}=\sum_{j=1}^{N-1}\sum_{\alpha =1}^{3}\frac{q_{eq,j}}{m_{re,j}}\frac{\partial }{\partial x_{j,\alpha }}D\left[M_{{\left(\left[1\right]\right)}_{j,\alpha },\left(\left[0\right]\right)}W_{\left([0]\right)}\right], \end{array}$$with *D* being a suitable approximation for the corresponding combination of quantum operator-continued fractions (omitting various dependences and indices in it) [32,34] and assuming approximately the initial condition $W_{in,([0])}$. This procedure and approximations are, in principle, independent of the specific properties of $\epsilon_{{\left(\left[2\right]\right)}_{j,\alpha ,l,\gamma }}$ (which displays quantum effects: compare with Section 6 and Section 7). The result in Equation (Equation 60) embodies stochasticity and displays a structure typical of diffusion-convection-reaction equations: it is linear, convection is determined by the *V*-dependent term and external sources are absent.

## 6. N = 2 Quantum Particles: Chemical Reactions

We shall apply the general formalism in Section 5 for *N*=2.

### 6.1. Assumptions on Two-Body Potential and Non-Equilibrium Moments and Hierarchy

Here: $V=v(|x_{1}-x_{2}|)$. Let us introduce the relative position vector $x=x_{2}-x_{1}$, the reduced mass *m* through $m^{-1}=m_{1}^{-1}+m_{2}^{-1}$ and $q_{eq}={\left(2m/\beta \right)}^{1/2}$.

(61)$$\begin{array}{c} H_{2,re}=-\frac{\hbar^{2}}{2m}\frac{\partial^{2}}{\partial x^{2}}+v\left(|x|\right). \end{array}$$

Spatial integrations over *x* is carried out inside the volume Ω.

Assumptions—The eigenfunctions corresponding to both $H_{CM}$ and $H_{2,re}$ will vanish, by assumption, at the surfaces enclosing Ω. We suppose that v(r) is an effective potential between two atoms or small molecules (which, in particular, includes and averages over Coulomb interactions). We shall suppose that such generic two-body potential v(r) satisfies the following conditions:v(r) is repulsive (>0) for $0\leq r<r_{0}$ (“hard core”, with adequately small $r_{0}$), attractive (<0) in the interval $r_{0}<r<{\left(3\Omega /4\pi \right)}^{1/3}$ and vanishes fast as $r\to {\left(3\Omega /4\pi \right)}^{1/3}$.v(r) is finite everywhere and its magnitude |v| is appreciable in $r_{0}<r<r_{0}+a<{\left(3\Omega /4\pi \right)}^{1/3}$. *a* is understood to be the range of *v*.v(r) and all $d^{n}v\left(r\right)/dr^{n}$, for n=1,2,3,…, are continuous for all r>0. Recall Section 4.1 and Section 5.1.v(r) does give rise to only one bound state (bound spectrum). Thus, the relevance of the region where v<0 is larger than that of the hard core.

Let $\varphi_{j}=\varphi_{j}\left(x\right)$ denote a suitably normalized eigenfunction of $H_{2,re}$ with corresponding eigenvalue $E_{j}$ and *j* denoting a set of labels. $H_{2,re}$ has both a discrete spectrum, corresponding to just one bound state and, above it and disjoint from it, to a denumerably infinite number of discrete states. For the bound state $E_{j}=E_{d}<0$ and $\varphi_{j}=\varphi_{d}\left(x\right)$. The denumerably infinite discrete spectrum (above the bound state) has a small spacing, and it becomes a continuous one (sweeping the continuous positive real axis) as $\Omega^{-1}\to 0$. We shall denote it by continuous spectrum (CS), both if either the small $\Omega^{-1}$ remains positive or if $\Omega^{-1}\to 0$. The eigenfunctions corresponding to the CS of $H_{2,re}$ are $\varphi_{j}=\varphi_{k}\left(x\right)$, with j≡k being an almost continuous wavevector, and eigenvalues $E_{j}=E_{k}\geq 0$. The CS eigenfunctions are normalized, if $\Omega^{-1}$ remains positive, through: $\left(\varphi_{k},\varphi_{k^{\prime }}\right)=\int d^{3}x\varphi_{k}^{*}\varphi_{k^{\prime }}=\delta_{kk^{\prime }}$ (a Kronecker delta). Also, $\left(\varphi_{d},\varphi_{d}\right)=\int d^{3}x\varphi_{d}^{*}\varphi_{d}=1$ (normalized) and $\left(\varphi_{d},\varphi_{k}\right)=\int d^{3}x\varphi_{d}^{*}\varphi_{k}=0$. Hence, $\varphi_{d}$ and all CS $\varphi_{k}$ span two separate Hilbert subspaces $H_{d}$ and $H_{CS}$. Let $\sum_{j}$ be a short-hand notation for summation (and, eventually, integration) which includes the contribution of both the bound state eigenfunction plus, if $\Omega^{-1}$ remains positive, that of a three-fold infinite summation over the whole CS. For the CS only: $\sum_{j}\to \left(\Omega /{\left(2\pi \right)}^{3}\right)\int d^{3}k$ as $\Omega^{-1}\to 0$. See [34]. Let $q=\left(-m_{2}q_{1}+m_{1}q_{2}\right)/\left(m_{1}+m_{2}\right)$ be the relative momentum vectors. The equilibrium Wigner function for the relative degrees of freedom reads: (62)$$\begin{array}{c} W_{2,re,eq}\left(x,q\right)=\frac{1}{{\left(\pi \hbar \right)}^{3}}\int d^{3}x^{\prime }\exp\left(\frac{i2qx^{\prime }}{\hbar }\right)\sum_{j}\exp\left[-\beta E_{j}\right]\varphi_{j}\left(x-x^{\prime }\right)\varphi_{j}^{*}\left(x+x^{\prime }\right). \end{array}$$

We shall consider the infinite family of orthogonal polynomials $H_{\left[n\right]}\left(y\right)$ generated by $W_{2,re,eq}$, as indicated in Section 5.2. We define $\left[n\right]=\left(n_{1},n_{2},n_{3}\right)$, with [0]=(0,0,0), ${\left[1\right]}_{\alpha }$ representing a vector with a 1 in the α position with the rest of the components being zero, and ${\left[2\right]}_{\alpha }$ similar to ${\left[1\right]}_{\alpha }$ but with a 2 in the α position. We shall also employ: $H_{{\left[1\right]}_{\alpha }}=H_{{\left[1\right]}_{\alpha }}\left(y\right)=y_{\alpha }$, and $H_{{\left[2\right]}_{\alpha ,\gamma }}=y_{\alpha }y_{\gamma }+\epsilon_{{\left[2\right]}_{\alpha ,\gamma }}$. $\epsilon_{{\left[2\right]}_{\alpha ,\gamma }}$ is given in Appendix A. The derivation of an infinite linear hierarchy for the non-equilibrium moments $W_{\left[n\right]}$ ($=\int d^{3}yH_{\left[n\right]}\left(y\right)W_{2,re}\left(x,y;t\right)$) is a particular case of that in Section 5.2. The initial condition $W_{in,[n]}$ for $W_{\left[n\right]}$ is obtained by replacing $W_{2,re}$ by $W_{2,re,in}$. The general (*t*-reversible) hierarchy for N=2 and any [n] is ($m_{re,j}=m$, $q_{eq,j}=q_{eq}$), as a result:(63)$$\begin{array}{c} \frac{\partial W_{\left[n\right]}}{\partial t}=-\sum_{\gamma =1}^{3}M_{\left[n\right],[{\left(n+1\right)}_{\gamma }]}W_{\left[{\left(n+1\right)}_{\gamma }\right]}-\sum_{\left[n^{\prime }\right]}M_{\left[n\right],\left[n-n^{\prime }\right]}W_{\left[n-n^{\prime }\right]}, \end{array}$$with
(64)$$\begin{array}{c} M_{\left[n\right],[{\left(n+1\right)}_{\gamma }]}W_{{\left[n+1\right]}_{\gamma }}\equiv \frac{q_{eq}}{m}\frac{\partial W_{{\left[n+1\right]}_{\gamma }}}{\partial x_{\gamma }}. \end{array}$$
(65)$$\begin{array}{c} M_{{\left[1\right]}_{\alpha },\left[0\right]}W_{\left[0\right]}=-\frac{q_{eq}}{m}\sum_{\gamma =1}^{3}\frac{\partial }{\partial x_{\gamma }}\left(\epsilon_{{\left[2\right]}_{\alpha ,\gamma }}W_{\left[0\right]}\right)+\frac{1}{q_{eq}}\frac{\partial v}{\partial x_{\alpha }}W_{\left[0\right]}. \end{array}$$

Bearing in mind that $W_{\left[n\right]}=W_{n_{1},n_{2},n_{3}}$, the subscript ${\left[n+1\right]}_{\gamma }$ in $W_{\left[{\left(n+1\right)}_{\gamma }\right]}$ denotes successively $W_{n_{1}+1,n_{2},n_{3}}$, $W_{n_{1},n_{2}+1,n_{3}}$ and $W_{n_{1},n_{2},n_{3}+1}$, and so on for the corresponding interpretation of the sum $\sum_{\left[n^{\prime }\right]}M_{\left[n\right],\left[n-n^{\prime }\right]}W_{\left[n-n^{\prime }\right]}$ in (Equation 63), which is performed as the non-negative integers in $\left[n^{\prime }\right]$ grow until $\sum_{\alpha =1}^{3}n_{\alpha }^{\prime }=\sum_{\alpha =1}^{3}n_{\alpha }$. We shall omit, for simplicity, the $M_{\left[n\right],\left[n-n^{\prime }\right]}$’s. By invoking rotational invariance, the operator $M_{{\left[1\right]}_{\alpha },\left[0\right]}$ can be expected to behave as a three-dimensional vector regarding the index α (=1,2,3). Equation (65) corrects Equation (23) in [34], by adding what may well be called an off-diagonal correction (namely $\sum_{\gamma =1,\gamma \neq \alpha }^{3}$). Equation (A2) for α=γ versus α≠γ and the fact that v=v(r) illustrate that due to their angular dependences (and upon eventual averaging or integrations over the angles involved in $x_{\alpha }$ and $x_{\gamma }$), the contributions of the off-diagonal correction are overcome, in practice, by those due to $\epsilon_{{\left[2\right]}_{\alpha ,\alpha }}$. Such off-diagonal correction does not alter the developments and conclusions in [34], which relied on the diagonal contribution, namely that with γ=α.

Let $W_{2,re}=W_{2,re,eq}$. By using the expression for $\epsilon_{{\left[2\right]}_{\alpha ,\gamma }}$, given in Appendix A, together with Equation (Equation 57), we obtain that

(66)$$\begin{array}{c} M_{{\left[1\right]}_{\alpha },\left[0\right]}W_{eq,[0]}=0. \end{array}$$

### 6.2. Chemical Reactions: Assumptions, Order of Magnitude Estimates and Approximations 

From now on, we shall consider values typical of microscopic scales and phenomena. The range *a* of *v* is a few Å (for instance, 3 up to 12 Å). Let $m=n\times m_{ne}$, with $m_{ne}$ being the neutron mass. We shall introduce the length scale δx=a/l, where l>2, characterizing approximately the smallest scale of appreciable variations for *v*. For instance, δx=(1/5)a, so that 0.6 Å<δx<2.5 Å. Moreover, the magnitude $v_{0}$ of |v|, averaged over the domain in which v<0, lies between 1 and 10 electronvolts. Notice that $v_{0}$ has a similar order of magnitude as typical energies $|E_{d}|$ for the bound state (see [5]). $\delta v(<v_{0})$ will denote the average variation of *v* within its range (where v≠0) in a scale δx. Similarly, $\delta^{n}v$, n=2,3,…, will denote the average variation of $\delta^{n-1}v$ in a scale δx ($\delta^{1}v\equiv \delta v$). For estimates, one could regard that |δv| be about one order of magnitude smaller than $v_{0}$ and that $|\delta^{n}v|$ be about one order of magnitude smaller than $|\delta^{n-1}v|$, n=2,3,...

Let $\lambda_{th}=\hbar {\left[\beta /2m\right]}^{1/2}=\hbar /q_{eq}$ be some suitable thermal wavelength. We shall assume that *n* and *T* are such that the following conditions are fulfilled: (a) $\lambda_{th}$ is smaller than δx, which sets a lower limit on nT; and (b) $\left(3/2\right)k_{B}T$ is smaller than or, at most, about $v_{0}$, which sets an upper limit on *T*. Both (a) and (b) correspond to the regime of typical chemical reactions (that is, a quantum regime but with small $\lambda_{th}$). Thus, even if the relative particle is in the quantum regime (but not in the classical high-temperature one), $\lambda_{th}$ is, on average, smaller than δx and *a*.

As zeroth-order approximations (see [34]), we shall take $\epsilon_{{\left[2\right]}_{\alpha ,\alpha }}≃-\left(1/2\right)$ for $|x|>a+r_{0}$ (dominated by the CS and, in turn, approximated through the classical Boltzmann equilibrium distribution and $\epsilon_{{\left[2\right]}_{\alpha ,\alpha }}≃-\left(\lambda_{th}^{2}/{\left(\delta x\right)}^{2}\right)$ for $r_{0}<\left|x\right|<a+r_{0}$ (dominated by the bound state), both independent of the index α and on the angular coordinates. The order of magnitude of $\epsilon_{{\left[2\right]}_{\alpha ,\gamma }}$, α≠γ can be estimated to be smaller than those of $\epsilon_{{\left[2\right]}_{\alpha ,\alpha }}$.

The order of magnitude involved upon applying $\left(q_{eq}/m\right)\left(\partial /\partial x_{\alpha }\right)$ (that is, $M_{\left[n\right],[{\left(n+1\right)}_{\alpha }]}$), is of the order of $\hbar /(\lambda_{th}m\delta x)$. The operator $M_{{\left[1\right]}_{\alpha },\left[0\right]}$ in Equation (65) gives rise to: (i) $\left(1/q_{eq}\right)\left(\partial v/\partial x_{\alpha }\right)$, which has an order of magnitude about $(\lambda_{th}/\hbar \delta x)\delta v$ for $r_{0}<\left|x\right|<r_{0}+a$ and fully negligible for $|x|>r_{0}+a$; (ii) $\left(q_{eq}/m\right)\epsilon_{{\left[2\right]}_{\alpha ,\alpha }}\left(\partial /\partial x_{\alpha }\right)$, which has an order of magnitude about $\epsilon_{{\left[2\right]}_{\alpha ,\alpha }}\hbar /\left(\lambda_{th}m\delta x\right)$ where, in turn, $\epsilon_{{\left[2\right]}_{\alpha ,\alpha }}$ has been estimated above for $r_{0}<\left|x\right|<r_{0}+a$ and for $|x|>r_{0}+a$; (iii) $\left(q_{eq}/m\right)\left(\partial \epsilon_{{\left[2\right]}_{\alpha ,\alpha }}/\partial x_{\alpha }\right)$, which has an order of magnitude of (at most) $\hbar /\left(\lambda_{th}m\delta x\right)\epsilon_{{\left[2\right]}_{\alpha ,\alpha }}$ for $r_{0}<\left|x\right|<r_{0}+a$, and negligible for $|x|>r_{0}+a$. In practice, the order of magnitude involved upon applying $M_{{\left[1\right]}_{\alpha },\left[0\right]}$ is the sum of (i) plus (ii) (since that of (iii) will be neglected).

The orders of magnitude of $M_{\left[n],[n-1\right]}$ can be expected to be respectively similar to the various contributions of $M_{{\left[1\right]}_{\alpha },\left[0\right]}$. On the other hand, the specific quantum contributions $M_{\left[n\right],\left[n-n^{\prime }\right]}$, $n^{\prime }>1$ have structures proportional to $\left(\hbar^{2n^{\prime \prime }}/q_{eq}^{2n^{\prime \prime }+1}\right)$ times spatial partial derivatives of *V* of order $2n^{\prime \prime }+1$ (for various $n^{\prime \prime }$, $2n^{\prime \prime }+1$ growing as $2^{-1}\sum_{\alpha =1}^{3}n_{\alpha }-1$). Such structures yield contributions smaller than that of $\left(1/q_{eq}\right)\left(\partial V/\partial x_{\alpha }\right)$ by factors ${\left(\lambda_{th}/\delta x\right)}^{2n^{\prime \prime }}\times \left(\delta^{2n^{\prime \prime }}V/\delta V\right)$, by virtue of assumption a) above. Moreover, one could expect $\left|(\delta^{2n^{\prime \prime }}v/\delta v)\right|$ to be smaller than unity and to decrease as $n^{\prime \prime }$ grows. The quantum contributions $M_{\left[n\right],\left[n-n^{\prime }\right]}$, $n^{\prime }>1$, will be neglected from now on, compared to that of $M_{\left[n],[n-1\right]}$. Under the above assumption a), the overall order of magnitude involved upon applying $M_{\left[n],[n-1\right]}$ in (Equation 44) is about the same as for $M_{{\left[1\right]}_{\alpha },\left[0\right]}$. Namely, the sum of (i) plus (ii) plus (iii). For further details, see [34].

### 6.3. Approximate Equation for Lowest Non-Equilibrium Moment

By using the above assumptions and estimates, one analyzes the non-equilibrium hierarchy Equation (Equation 63) for the $W_{\left[n\right]}$’s, and proceeds to the small thermal wavelength quantum regime. In the latter, *x* varies by units of order $\delta x>\lambda_{th}$. Then, by neglecting all $M_{\left[n\right],{\left[n-n^{\prime }\right]}_{\gamma }}$’s with $\sum_{\alpha =1}^{3}n_{\alpha }^{\prime }>1$, Equation (Equation 63) becomes, as a result, the approximate (*t*-reversible) three-term hierarchy (actually, the N=2 counterpart of Equation (Equation 59)):(67)$$\begin{array}{c} \frac{\partial W_{\left[n\right]}}{\partial t}=-\sum_{\gamma =1}^{3}M_{\left[n\right],[{\left(n+1\right)}_{\gamma }]}W_{\left[{\left(n+1\right)}_{\gamma }\right]}-\sum_{\gamma =1}^{3}M_{\left[n],[n-1\right]}W_{\left[n-1\right]}. \end{array}$$

Assumption and approximation—We shall assume the initial condition $W_{in,[0]}\neq 0$ and $W_{in,[n]}=0$ for [n]≠[0] We solve the approximate Equation (Equation 67) and proceed to the long-term approximation as in Section 5.3. One gets the following irreversible approximate quantum (Smoluchowski-like) equation for the lowest non-equilibrium moment $W_{\left[0\right]}\left(x;t\right)$:(68)$$\begin{array}{c} \frac{\partial W_{\left[0\right]}}{\partial t}=\frac{q_{eq}}{m}\sum_{\alpha =1}^{3}\frac{\partial }{\partial x_{\alpha }}D\left[M_{{\left[1\right]}_{\alpha },\left[0\right]}W_{\left[0\right]}\right], \end{array}$$this result (with a constant *D*) is the N=2 counterpart of Equation (Equation 60). At a later stage, the constant *D* could be supposed to be positive. See [34] for a detailed analysis and applications to a binary chemical reaction. To the best of our knowledge, no general agreement has been reached to define a unique non-equilibrium chemical potential based on non-equilibrium statistical mechanics, so far [20]. Within such a limitation, approximate chemical equilibrium and kinetic equations for binary reactions have been analyzed in [34]. For binary chemical reactions in the quantum domain, mean first passage methods enabled to study approximately the times required in the transitions between bound state and unbound configurations and to exhibit typical Arrhenius exponential factors [34].

### 6.4. Comparison: Binary Chemical Reactions in Classical Statistical Mechanics 

We focus on the classical statistical evolution of the relative degrees of freedom, described by the relative position vector $x=x_{2}-x_{1}$ and momentum ***q*** and by $W_{c}\left(x,q;t\right)$ (with factored out the CM). Here we shall work with the moments $W_{c,re}\left(\left[n\right]\right)=W_{c,re}\left(x;\left[n\right];t\right)$ for the relative degrees of freedom (equivalent to and more advantageous than the g([n])’s in Section 3 for dealing with chemical reactions).

Assumption and approximation—We shall assume the initial condition $W_{c,re,in}\left(\left[0\right]\right)\neq 0$ and $W_{c,re,in}\left(\left[n\right]\right)=0$ for [n]≠[0]. The developments in Section 3 can be directly applied to the present case and yield an infinite hierarchy for all moments $W_{c,re}\left(\left[n\right]\right)$. The long-term approximation for the g([n])’s can be recast as a corresponding one for the $W_{c,re}\left(\left[n\right]\right)$.

We arrive, as a result, at the approximate irreversible equation for the lowest non-equilibrium moment $W_{c,re}\left(\left[0\right]\right)$, with initial condition $W_{c,in}\left(\left[0\right]\right)$: (69)$$\begin{array}{c} \frac{\partial W_{c,re}\left(\left[0\right]\right)}{\partial t}=\frac{q_{eq}}{m}\sum_{\alpha =1}^{3}\frac{\partial }{\partial x_{\alpha }}D\left[M_{{\left[1\right]}_{\alpha },\left[0\right],c}W_{c,re}\left(\left[0\right]\right)\right] \end{array}$$(70)$$\begin{array}{c} M_{{\left[1\right]}_{\alpha },\left[0\right],c}W_{c,re}\left(\left[0\right]\right)=\frac{q_{eq}}{2m}\frac{\partial }{\partial x_{\alpha }}\left(W_{c,re}\left(\left[0\right]\right)\right)+\frac{1}{q_{eq}}\frac{\partial v}{\partial x_{\alpha }}W_{c,re}\left(\left[0\right]\right). \end{array}$$

The main differences between Equations (65) and (70) are the vanishing of $\epsilon_{{\left[2\right]}_{\alpha ,\gamma }}$ for α≠γ and the replacement of $-\epsilon_{{\left[2\right]}_{\alpha ,\alpha }}$ by 1/2, so that $M_{{\left[1\right]}_{\alpha },\left[0\right]}$ becomes $M_{{\left[1\right]}_{\alpha },\left[0\right],c}$. The equilibrium solution $W_{c,re,eq}\left(\left[0\right]\right)$ ($M_{{\left[1\right]}_{\alpha },\left[0\right],c}W_{c,re,eq}\left(\left[0\right]\right)=0$) is proportional to exp(−βV). Equation (Equation 69) is a standard classical diffusion equation: so, we have rederived Equation (7.30) in [55]. The interest of the latter equation stems from the fact that for suitable *v* (for instance, the well-known Morse potential) and even in the absence of explicit quantum effects, it has been used as a zeroth-order model for a binary chemical reaction. Specifically, it has enabled to estimate the recombination rate of a diffusing molecule in three spatial dimensions, through mean first-time passage methods [55].

## 7. N = 3 Quantum Particles: Chemical Reactions

We shall apply the general formalism in Section 5 for *N* = 3.

### 7.1. Some General Aspects

Assumptions—Here: (71)$$\begin{array}{c} V=v_{12}\left(\right|x_{2}-x_{1}\left|\right)+v_{23}\left(\right|x_{3}-x_{2}\left|\right)+v_{31}\left(\right|x_{1}-x_{3}\left|\right) \end{array}$$

We assume that any of the two-body potentials in Equation (Equation 71) fulfill the same assumptions as v(r) in Section 6.1 (*r* denoting here the distance between any two particles, say, any of $|x_{2}-x_{1}|$, $|x_{3}-x_{2}|$, $|x_{1}-x_{3}|$). Then, the two particle 1 and 2 have one bound state with energy $E_{d(12)}\left(<0\right)$ and so on for 2 and 3 and for 3 and 1. We assume $E_{d(12)}<E_{d(23)}<E_{d(31)}\left(<0\right)$. For simplicity, we shall assume that there are no bound states of the three particles.

We shall also factor out the CM. We have three natural choices or options for the relative degrees of freedom, denoted as ((12),3), ((23),1) and ((31),2). See Appendix B for the corresponding relative position vectors and momenta and for several useful formulae employed in our study, without further reference to them. We shall consider the ((12),3) choice. Then, $H_{3,re}$ becomes: (72)$$\begin{array}{c} H_{3,re}=-\frac{\hbar^{2}}{2m_{12}}\frac{\partial^{2}}{\partial x_{12}^{2}}-\frac{\hbar^{2}}{2m_{3(12)}}\frac{\partial^{2}}{\partial x_{3(12)}^{2}}+V\left(\left(x\right)\right). \end{array}$$and so on for the representations of $H_{3,re}$ in terms of the other two choices. Having discussed in Section 6.1 the spectrum of $H_{2,re}$ for small $\Omega^{-1}$ and for $\Omega^{-1}\to 0$, we shall outline the spectrum of $H_{3,re}$ directly for $\Omega^{-1}\to 0$. General studies on the quantum-mechanical three-body problem [56] imply that $H_{3,re}$ has: (i) an infinite CS (i)) in $\left(E_{d(12)},+\infty \right)$, associated with the choice ((12),3), corresponding to particle 3 interacting with the bound state of (1,2), so that the former could be far from the latter, other two similar CS corresponding respectively to the choices (ii) ((23),1) (CS (ii)) and (iii) ((31),2) (CS (iii)), and (iv) an infinite CS (CS (iv)) in (0,+∞) corresponding to the three unbound particles, so that any of them could be far from the other two, but still interacting among themselves. Thus, CS (ii) corresponds to particle 1 interacting with the bound state of (2, 3), so that the former could be far from the latter and so on for CS (iii). Here, $\sum_{j}$ will denote a short-hand notation for summations and integrations over the whole 4 infinite CS’s in (i), (ii), (iii) and (iv) above. Below, $\varphi_{j}=\varphi_{j}\left(x_{12},x_{3(12)}\right)$ will denote an eigenfunction of $H_{3,re}$ (suitably normalized as $\Omega^{-1}\to 0$, in analogy with Section 6.1) with continuous eigenvalue $E_{j}$. The equilibrium Wigner function for the relative degrees of freedom in the ((12),3) choice reads: (73)$$\begin{array}{c} W_{3,re,eq}\left(x_{12},x_{3(12)},q_{12},q_{3(12)}\right)=\frac{1}{{\left(\pi \hbar \right)}^{6}}\int d^{3}x_{12}^{\prime }d^{3}x_{3(12)}^{\prime }\exp\left(\frac{2i(q_{12}x_{12}^{\prime }+q_{3(12)}x_{3(12)}^{\prime })}{\hbar }\right)\times \\ \sum_{j}\exp\left[-\beta E_{j}\right]\varphi_{j}\left(x_{12}-x_{12}^{\prime },x_{3(12)}-x_{3(12)}^{\prime }\right)\varphi_{j}^{*}\left(x_{12}+x_{12}^{\prime },x_{3(12)}+x_{3(12)}^{\prime }\right). \end{array}$$$q_{12}$ and $q_{3(12)}$ being defined in Appendix B, and so on for the other two choices. Notice that $W_{3,re,eq}$ is the same for the three choices.

### 7.2. Approximate Equation for Lowest Non-Equilibrium Moment

Here, $q_{eq,1}=q_{eq,12}={\left(2m_{12}/\beta \right)}^{1/2}$ and $q_{eq,2}=q_{eq,3(12)}={\left(2m_{3(12)}/\beta \right)}^{1/2}$.

Assumptions and approximations—For N=3, we shall follow the general strategy in Section 5.2 and Section 5.3 and make assumptions similar to those for N=2 in Section 6.2. Then, the estimates in Section 6.2 can be readily extended for N=3, $q_{eq}$ being now replaced by $q_{eq,12}$ and $q_{eq,3(12)}$ (all of which with more or less similar orders of magnitude) and *v* denoting now any of the two-body potentials (no estimates being given explicitly for $\epsilon_{{\left(\left[2\right]\right)}_{j,\alpha ,l,\gamma }}$, in our short analysis). We shall consider the regime of small thermal wavelength and the long-term approximation.

Then, as a result, one gets the following approximate irreversible quantum (Smoluchowski-like) equation for the lowest non-equilibrium moment $W_{\left([0]\right)}\left(x_{12}x_{3(12)};t\right)$ for the relative degrees of freedom in the ((12), 3) choice: ($x_{12}=\left(x_{12,\alpha }\right)$, $x_{3(12)}=\left(x_{3(12),\alpha }\right)$, α=1,2,3): (74)$$\begin{array}{c} \frac{\partial W_{\left([0]\right)}}{\partial t}=\frac{q_{eq,12}}{m_{12}}\sum_{\alpha =1}^{3}\frac{\partial }{\partial x_{12,\alpha }}D_{12}\left[M_{{\left(\left[1\right]\right)}_{12,\alpha },\left(\left[0\right]\right)}W_{\left([0]\right)}\right]+\\ \frac{q_{eq,3(12)}}{m_{3(12)}}\sum_{\alpha =1}^{3}\frac{\partial }{\partial x_{3(12),\alpha }}D_{3(12)}\left[M_{{\left(\left[1\right]\right)}_{3(12),\alpha },\left(\left[0\right]\right)}W_{\left([0]\right)}\right] \end{array}$$(75)$$\begin{array}{c} M_{{\left(\left[1\right]\right)}_{12,\alpha },\left(\left[0\right]\right)}W_{\left([0]\right)}=-\frac{q_{eq,12}}{m_{12}}\sum_{\gamma =1}^{3}\frac{\partial }{\partial x_{12,\gamma }}\left(\epsilon_{{\left(\left[2\right]\right)}_{12,\alpha ,12,\gamma }}W_{\left([0]\right)}\right)\\ -\frac{q_{eq,3(12)}}{m_{3(12)}}\sum_{\gamma =1}^{3}\frac{\partial }{\partial x_{3(12),\gamma }}\left(\epsilon_{{\left(\left[2\right]\right)}_{12,\alpha ,3(12),\gamma }}W_{\left([0]\right)}\right)+\frac{1}{q_{eq,12}}\frac{\partial V}{\partial x_{12,\alpha }}W_{\left([0]\right)} \end{array}$$(76)$$\begin{array}{c} M_{{\left(\left[1\right]\right)}_{3(12),\alpha },\left(\left[0\right]\right)}W_{\left([0]\right)}=-\frac{q_{eq,12}}{m_{12}}\sum_{\gamma =1}^{3}\frac{\partial }{\partial x_{12,\gamma }}\left(\epsilon_{{\left(\left[2\right]\right)}_{3(12),\alpha ,12,\gamma }}W_{\left([0]\right)}\right)\\ -\frac{q_{eq,3(12)}}{m_{3(12)}}\sum_{\gamma =1}^{3}\frac{\partial }{\partial x_{3(12),\gamma }}\left(\epsilon_{{\left(\left[2\right]\right)}_{3(12),\alpha ,3(12),\gamma }}W_{\left([0]\right)}\right)+\frac{1}{q_{eq,3(12)}}\frac{\partial V}{\partial x_{3(12),\alpha }}W_{\left([0]\right)} \end{array}$$with the initial condition $W_{in,([0])}\neq 0$ ($W_{in,([n])}=0$ for ([n])≠([0])). The ϵ’s are defined in Appendix B and obtained near the classical regime (to next to leading order). We have approximated the corresponding combinations of quantum operator-continued fractions by the constants $D_{12}$ and $D_{3(12)}$. This procedure and approximations are, in principle, independent of the specific properties of the ϵ’s. Notice that $W_{\left([0]\right)}$ is the same for the three choices ((12),3)
((23),1) and ((31),2), by virtue of Equation (A23). On the other hand, this unique $W_{\left([0]\right)}$ satisfies other two equations similar to Equation (74), for the other two choices ((23),1) and ((31),2).

Notice that for $W_{3,re}=W_{3,re,eq}$, the corresponding moment $W_{eq,([0])}$ fulfills both exact equations $M_{{\left(\left[1\right]\right)}_{12,\alpha },\left(\left[0\right]\right)}W_{eq,([0])}=0$ and $M_{{\left(\left[1\right]\right)}_{3(12),\alpha },\left(\left[0\right]\right)}W_{\mathrm{eq},([0])}=0$, which generalize Equation (Equation 66). Suitable combinations of the last two exact equations for the ((12),3) choice have to transform into the corresponding exact two equations for each of the ((23), 1) and ((31), 2 choices): there are three equivalent pairs of equations for the same $W_{eq,([0])}$. We omit details.

### 7.3. Comparison: N=3 Particles in Classical Statistical Mechanics

We shall treat the N=3 counterpart of Equations (Equation 69) and (70), for the classical $W_{c,re}\left(\left[0\right]\right)$, for the relative degrees of freedom. Through the corresponding assumptions and approximations, we get as a result, for the ((12), 3) choice: (77)$$\begin{array}{c} \frac{\partial W_{c,re}\left(\left(\left[0\right]\right)\right)}{\partial t}=\frac{q_{eq,12}}{m_{12}}\sum_{\alpha =1}^{3}\frac{\partial }{\partial x_{12,\alpha }}D_{12,c}\left[M_{{\left(\left[1\right]\right)}_{12,\alpha },\left(\left[0\right]\right),c}W_{c,re}\left(\left(\left[0\right]\right)\right)\right]+\\ \frac{q_{eq,3(12)}}{m_{3(12)}}\sum_{\alpha =1}^{3}\frac{\partial }{\partial x_{3(12),\alpha }}D_{3(12)c}\left[M_{{\left(\left[1\right]\right)}_{3(12),\alpha },\left(\left[0\right]\right),c}W_{c,re}\left(\left(\left[0\right]\right)\right)\right] \end{array}$$(78)$$\begin{array}{c} M_{{\left(\left[1\right]\right)}_{12,\alpha },\left(\left[0\right]\right),c}W_{c,re}\left(\left(\left[0\right]\right)\right)=\frac{q_{eq,12}}{2m_{12}}\frac{\partial }{\partial x_{12,\alpha }}\left(W_{c,re}\left(\left(\left[0\right]\right)\right)\right)+\frac{1}{q_{eq,12}}\frac{\partial V}{\partial x_{12,\alpha }}W_{c,re}\left(\left(\left[0\right]\right)\right) \end{array}$$
(79)$$\begin{array}{c} M_{{\left(\left[1\right]\right)}_{3(12),\alpha },\left(\left[0\right]\right),c}W_{c,re}\left(\left(\left[0\right]\right)\right)=\frac{q_{eq,3(12)}}{2m_{3(12)}}\frac{\partial }{\partial x_{3(12),\alpha }}\left(W_{c,re}\left(\left(\left[0\right]\right)\right)\right)+\frac{1}{q_{eq,3(12)}}\frac{\partial V}{\partial x_{3(12),\alpha }}W_{c,re}\left(\left(\left[0\right]\right)\right) \end{array}$$

These turn out to come from Equations (74)–(76) with simplifications in the classical regime, namely, $-\epsilon_{{\left(\left[2\right]\right)}_{12,\alpha ,12,\alpha }}$ and $-\epsilon_{{\left(\left[2\right]\right)}_{3(12),\alpha ,3(12),\alpha }}$ are approximated by 1/2, while all the remaining $-\epsilon_{{\left(\left[2\right]\right)}_{j,\alpha ,l,\gamma }}$’s vanish. We have approximated the corresponding constants $D_{12}$ and $D_{3(12)}$ by the constants $D_{12,c}$ and $D_{3(12),c}$ (to be supposed positive, eventually).

Let us also consider, avoiding writing them in detail: (i) the *t*-dependent counterpart of Equation (77) for the ((23), 1) choice, with corresponding constants $D_{23,c}$ and $D_{1(23),c}$, and (ii) the *t*-dependent counterpart of Equation (77) for the ((31), 2) choice, with corresponding constants $D_{31,c}$ and $D_{2(31),c}$.

Assumption—We shall complete the approximation by assuming that:(80)$$\begin{array}{c} D_{12,c}=D_{3(12),c}=D_{23,c}=D_{1(23),c}=D_{31,c}=D_{2(31),c} \end{array}$$

Then, the following results follow: (1) by using Equations (A21) and (A22), it follows that the above *t*-dependent Equation (77) for the ((12), 3) choice transforms exactly into the corresponding equation for the ((23), 1) choice and, similarly; (2) the *t*-dependent Equation (77) for the ((12), 3) choice transforms exactly into the corresponding equation for the ((31), 2) choice.

It is natural to conjecture that the *t*-dependent quantum Equations (74)–(76) for the ((12), 3) choice transform exactly into the corresponding (also *t*-dependent) quantum equations for the ((23), 1) and ((31), 2) choices, by using assumptions regarding the quantum coefficients $D_{12}$, $D_{3(12)}$ and the corresponding $D_{23}$, $D_{1(23)}$, $D_{31}$ and $D_{2(31)}$ for the other two choices. The latter assumptions would be the counterparts of the classical ones implemented through Equation (80). The derivations necessary to implement such a conjecture are not transparent and, so, they will lie out of our scope here.

The equilibrium solution of Equation (77) is proportional to $\exp[-{\left(k_{B}T\right)}^{-1}\left(v_{12}+v_{23}+v_{31}\right)]$. One arrives at the same equilibrium solution by using the counterparts of Equation (77) for the other two choices.

It may well be that the classical Equation (77), for the three different choices and for suitable $v_{12}+v_{23}+v_{31}$, could be useful as a zeroth-order model to estimate quantities characterizing specific transitions among the different particles, as its classical N=2 counterpart (Equation 69) did for N=2 reactions [55]. A study of this possibility for N=3 lies outside our scope here.

### 7.4. Thermal and Chemical Equilibria: An Approximate (Semi-Quantitative) Discussion

Notice that no chemical potentials have been introduced, which does not appear to be straightforward in our Wigner function framework. The semi-quantitative discussion below will try to provide some bypass to discuss chemical equilibrium in a semi-quantitative way.

Let us now return to Equation (73). Upon integration we get ($y_{12}=q_{eq,12}^{-1}q_{12}$, $y_{3(12)}=q_{eq,3(12)}^{-1}q_{3(12)}$)
(81)$$\begin{array}{c} \int d^{3}x_{12}d^{3}x_{3(12)}d^{3}y_{12}d^{3}y_{3(12)}W_{3,re,eq}\left(x_{12},x_{3(12)},q_{12},q_{3(12)}\right)=\\ {\left(\frac{1}{q_{eq,12}q_{eq,3(12)}}\right)}^{3}\sum_{j}\exp\left[-\beta E_{j}\right]\int d^{3}x_{12}d^{3}x_{3(12)}|\varphi_{j}\left(x_{12},x_{3(12)}\right){|}^{2} \end{array}$$

Notice that $\int d^{3}x_{12}$ and $\int d^{3}x_{3(12)}$ are extended over the volume Ω (microscopically large but macroscopically small).

For sufficiently low $k_{B}T$, the contribution of the CS (i) is the dominant one to Equation (81). As $k_{B}T$, varying initially in the range where only the system formed by particle 3 interacting with the bound state of (1, 2) is possible (lower part of the CS (i)), increases and approaches the range where the system formed by particle 1 interacting with the bound state of (2, 3) is also possible (CS (ii)), the transitions (chemical reactions) (12)+3→(23)+1 and its inverse (23)+1→(12)+3 occur. As such $k_{B}T$’s, other chemical reactions involving the states corresponding to CS (iii) and CS (iv) are not produced. However, transitions involving the states corresponding to CS (iii) and CS (iv) are indeed possible as $k_{B}T$ continues to increase.

We shall focus on $k_{B}T$ś such that only (12)+3→(23)+1 and its inverse (23)+1→(12)+3 occur (1 and 2 being bound to each other for (12) and 2 and 3 being bound to each other for (23)). Then, the dominant contributions to (81) come from such states in CS (i) and CS (ii) (the contributions from CS (iii) and CS (iv) being exponentially damped compared to the former).

The above two sets of states (12)+3 and (23)+1 in the CS (i) and CS (ii) correspond to thermal equilibrium, but there may be, in general, an imbalance between their contributions to $W_{3,re,eq}$, unless some further requirement be fulfilled. To grasp the latter, in a semi-quantitative framework, we advance that there exist key ranges of intermediate *T*ś and volumes (microscopically large but macroscopically small) such that the orders of magnitude of the contributions of both sets (12)+3 and (23)+1 are not significantly different from each other at thermal equilibrium. The order of magnitude of the contribution of the states (12)+3 to $W_{3,re,eq}$ in Equation (81) will be estimated, as a zeroth-order approximation, by approximating it as the product of the bound state wave function for the pair (12) ($\varphi_{d(12)}(x_{12}$)) times the classical equilibrium distribution for particle 3 treated as a free particle with any momentum $q_{3(12)}$ (thereby omitting the interaction of the bound state of (1, 2) with 3). Accordingly, we must integrate over $x_{12}$, $x_{3(12)}$ and $q_{3(12)}$. We get: (82)$$\begin{array}{c} {\left(\frac{1}{q_{eq,12}q_{eq,3(12)}}\right)}^{3}\exp\left(-\beta E_{d(12)}\right)\int d^{3}x_{12}{\left|\varphi_{d(12)}\left(x_{12}\right)\right|}^{2}\int d^{3}x_{3(12)}\times \\ \int d^{3}q_{3(12)}\frac{1}{{\left(2\pi \right)}^{3}}\exp\left[-\beta \frac{q_{3(12)}^{2}}{2m_{3(12)}}\right]≃{\left(\frac{1}{q_{eq,12}q_{eq,3(12)}}\right)}^{3}\exp\left(-\beta E_{d(12)}\right)\frac{\Omega_{3}q_{eq,3(12)}^{3}}{8\pi^{3/2}\hbar^{3}}. \end{array}$$

The effective integration region for $x_{12}$ should be smaller than $\Omega_{3}$. Using the equivalent counterpart of (73) for the choice ((23), 1), a similar estimate for the bound state of (2, 3) times the classical equilibrium distribution for particle 1, treated as free, yields:(83)$$\begin{array}{c} {\left(\frac{1}{q_{eq,23}q_{eq,1(23)}}\right)}^{3}\exp\left(-\beta E_{d(23)}\right)\frac{\Omega_{1}q_{eq,1(23)}^{3}}{8\pi^{3/2}\hbar^{3}}. \end{array}$$

For chemical equilibrium in the transition under study, it seems natural that there be a (at least semi-quantitative) similarity of the orders of magnitude of both Equations (82) and (83). That could be possible, for the given $E_{d(12)}$ and $E_{d(23)}$ and masses involved, for suitable $\Omega_{1}$, $\Omega_{3}$, for the *T*’s under consideration.

We shall now discuss the size of the microscopically large Ω and, hence, that of $\Omega_{1}$ and $\Omega_{3}$. One could argue physically that our three-particle system is contained in some macroscopic gaseous system at *T*, containing a very large number *N* of similarly interacting triplets (upon neglecting the interactions among different or disjoint triplets). That is, our three-particle system is just one of those triplets. The gaseous system occupies a macroscopically large volume $\Omega^{\prime }\gg \Omega$. Then, we argue that $\Omega ≃\Omega^{\prime }/N$ and that the microscopically large $\Omega_{1}$ and $\Omega_{3}$ could be possibly somewhat smaller than Ω. Notice that $\Omega_{3}$ could be regarded as a (microscopically large but macroscopically small) allowed volume for particle 3, per triplet ((12), 3) when (12) are in a bound state in the gaseous system and so on for $\Omega_{1}$ in a corresponding triplet ((23), 1). Such $\Omega_{1}$ and $\Omega_{3}$ could allow for the compatibility between the orders of magnitude in Equations (82) and (83), for the *T*’s considered. For a comparison, recall that typical diatomic gases at adequately low pressures and *T*’s not smaller than room temperature, one has that Ω is about $10^{4}$–$10^{5}$ Å${}^{3}$ per pair [4].

Then, when such a (at least semi-quantitative) similarity of the orders of magnitude occurs, there could be simultaneous coexistence of thermal and chemical equilibria for (12)+3→(23)+1 and its inverse (23)+1→(12)+3, in the actual simple framework.

## 8. Conclusions and Discussion

In short, and to avoid repetitions, the program in this work is the following in all cases. By assumption, the system evolves subject to a hb, at equilibrium at absolute temperature T, in all cases, except in Section 3 (in which a large closed system will be treated). Ab initio dissipation is excluded, by assumption, in all cases. We omit any analysis of the states of the hb and, in an effective way, we focus exclusively on the states of the system considered. The analysis is based upon the non-equilibrium classical Liouville probability distribution function $W_{c}$ and the quantum Wigner function *W*. Extensive use is made of the corresponding equilibrium functions $W_{c,eq}$ and $W_{eq}$ as weight functions to generate orthogonal polynomials. In turn, the latter are employed to construct non-equilibrium moments of the corresponding non-equilibrium functions, which fulfill non-equilibrium hierarchies (containing coefficients determined by the corresponding equilibrium functions) and with suitable initial conditions. Various approximate non-equilibrium classical and quantum-mechanical Smoluchowski-like equations are obtained for the lowest non-equilibrium moment for long time: they yield evolutions towards thermal equilibrium. Below, we shall emphasize distinct features in each case.

1. Reviewed material.

For the sake of simpler presentations, the various approximations leading to various Smoluchowski-like equations for lowest non-equilibrium moments are outlined for one-dimensional cases in Section 2 and Section 4, in the classical and quantum domains, respectively.

Classical case.

Section 2: classical one-dimensional particle. Here, $W_{c,eq}$ is Gaussian and the orthogonal polynomials are the standard Hermite ones. They are used to construct moments of the non-equilibrium $W_{c}$. In turn, those moments fulfill non-equilibrium hierarchies, the solutions of which depend on certain operator-continued fractions and combinations thereof. Under suitable long-term approximations performed on those operator-continued fractions, the lowest moment fulfills an irreversible Smoluchowski-like equation.

Section 3: classical closed N-particle system, in three dimensions, without hb. A specific analysis is devoted to the initial non-equilibrium state (describing equilibrium at large distances but off-equilibrium at finite distances). The analysis in Section 2 is suitably generalized. The non-equilibrium hierarchy is genuinely different from the non-equilibrium classical BBGKY hierarchy [7,8]. As an extreme approximation, an irreversible Smoluchowski-like equation (resembling that in the standard Rouse model for polymer dynamics [38]) is obtained. A non-increasing Liapunov function characterizes the long-term evolution towards equilibrium.

Quantum case.

Section 4: one-dimensional particle. Difficulties related to the non-positivity of *W* are bypassed, by invoking a suitable extension of the theory of orthogonal polynomials [54]. Based upon the latter, the equilibrium Wigner function is used to generate orthogonal polynomials which, in turn, lead to non-equilibrium moments. The corresponding development in Section 2 regarding evolutions is extended for the lowest non-equilibrium moment.

Section 6: various general developments in Section 5 (containing new material, as discussed below) are treated specifically for N=2 particles. On the other hand, developments in Section 4 are generalized non-trivially to this case. The resulting construction is applied to binary chemical reactions in the quantum domain, with applications for adequately small thermal wavelength. The corresponding classical Smoluchowski-like equation coincides with Equation (7.30) in [55], which has been a useful zeroth-order model for binary chemical reactions.

2. New material.

Classical case.

Section 2.4: an approximation for an operator-continued fraction in non-equilibrium classical statistical mechanics is proposed, in outline.

Quantum case.

Section 5: general formulation for N>1 quantum particles. It is a general extension, for *N* particles, of the case of binary chemical reactions, previously published in [34]. The statistical evolutions of the center of mass and of the relative degrees of freedom are factored out. An approximate irreversible quantum (Smoluchowski-like) equation for the lowest non-equilibrium moment describing the relative degrees of freedom is derived from the Wigner functions *W*.

Section 7: various general developments in Section 5 are treated specifically for N=3 particles in three spatial dimensions. Non-equilibrium chemical reactions involving three particles are analyzed, in typical regimes of short thermal wavelength and long times. Special attention is paid to the different possible choices of variables for the relative degrees of freedom. Based upon the developments in Section 5, an approximate irreversible quantum (Smoluchowski-like) equation for the lowest non-equilibrium moment is obtained. A semi-quantitative analysis of chemical equilibrium involving transitions among three particles is outlined. The Smoluchowski-like equations for chemical reactions involving three particles are analyzed in the classical domain, thereby justifying paying due attention, for the sake of consistency, to the different possible choices of variables for the relative degrees of freedom.

Some open problems deserving further research—We may quote: (op1) the classical three-term hierarchies and the associated operator-continued fractions, (op2) the quantum hierarchies (no longer three-term ones) and the associated operator solutions, (op3) the various approximations regarding (op1) and (op2) carried out throughout this work, (op4) chemical reactions in the quantum domain involving two and, specifically, three particles. In particular, it is suggested that the classical Equation (77) could be useful as a zeroth-order model for specific transitions among three reacting particles, as its classical N=2 counterpart in (Equation 69) did for N=2.

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
