# Peer review of "Non-Equilibrium Liouville and Wigner Equations: Classical Statistical Mechanics and Chemical Reactions for Long Times"

_entropy, 2019, doi:10.3390/e21020179_

Round 1

Reviewer 1 Report

The paper  offers a very  systematic combination of a  review material
used to outline  novel results and analysis in the  field of classical
and quantum  statistical mechanics. The general  process considered is
relaxation of systems towards thermodynamic equilibrium. The classical
and  quantum  (Wigner)  equilibrium  functions are  used  to  generate
families of  orthogonal polynomials further used  to construct moments
of the nonequilibrium counterparts of these functions. Irreversibility
and entropy are analyzed  under certain long-time approximations which
yield evolution towards thermal  equilibrium. Single and many particle
systems  and  systems  with  chemical  reactions  are  considered  and
analyzed.  The  logical  presentation  is  pedantic  and  didactically
constructed.  Unfortunately  this is  related only to  the specialized
text. The quality  of the English is bad, some  (actually most) of the
sentences e.g. in the Introduction look  like an outcome of the Google
Translator.  A check of all sentences is recommended.

Some corrections are suggested:
-Eqn. 12 and around - please give definition of s+ and z+, it is shorter than
to write (in standard notations 43).
-Page 11 top: The second sentence is incomplete.
-Page 13, the introduction of Eqn. 39 needs more explanations and in particular
a definition of \epsilon_{Q,n,n-j}
-The same holds for Eqn. 56.

Author Response

Manuscript ID: Entropy-405128
Reviewer 1
Response (January 16, 2019)

Dear Sir/Lady.

Thanks for your comments.

I have taken into account all your criticisms. I have also taken into account other criticisms.

Accordingly, I have carried out  the following  corrections    in the present revised version.

1.-  I have checked and modified somewhat the Introduction, in order to improve it.
Specifically, by following your recommendation, I have modified the various paragraphs in it,   revised the English and displaced a number of references from it to  Section 4.
Moreover, without altering the structure, ordering,  construction and equations of the work,   I have   introduced   throughout the whole text the  words “assumption”, “approximation”, “simplification” and “result”, as they become suitable. In this way,    the results and the assumptions, approximations and simplification under which the former  hold are clearly visible and separated from the actual derivations. This improvement in presentation is announced briefly  in two  lines at the end of the Introduction.

Also, in order that    what  is literature review  and what is the introduction of novel material be displayed and their differences be clear, I have   modified almost completely Section 8 (Conclusions and discussion).

And, in so doing, I have also revised the whole manuscript, as an attempt  to improve it.

 2.- The four  corrections that you have  suggested have been carried out, about the following equations (using the  numbering of the revised version)
-Eqs. (13) and (14) (formerly, Eqs. 12 and around, in connection with    standard notations for continued fractions in reference by W. Gautschi).
-Eqs. (32) and (33) (formerly, Page 11 top:  above former Eq. (31)).
-Eq. (41)  (formerly,  Eqn. (39) , regarding  the
 definition of \epsilon_{Q,n,n-j}
-Eqs. (58) and (59) (formerly Eq. (56)).

3. A new correction has been performed, embodied in the new Eq (81). And a new equation (Eq. (106)) has been added, in order to clarify the justification of a result in Section 7.3.

4) References have been reordered, as commented above, and two new references have been added, for completeness, namely the new references 16 and 56.

Reviewer 2 Report

This paper deals with classical and quantum interacting many particle systems which are in non-equilibrium and subject to a thermal heat bath. The particles evolve according to the Liouville equation (classical case) resp. the Wigner equation (quantum case). The technique used is an expansion in terms of orthogonal polynomials which are used to construct moments of the probability distribution function describing the positions and momenta of the particles. Those moments fulfill infinite hierarchies, and under certain conditions the lowest moment fulfils an irreversible Smoluchowski-like equation.

The paper reads like a quite long derivation, and although the calculations seem sound, it is unfortunately never quite clear to what end they are performed. The author should revise the paper in such a way that its intend is clearly stated and the new contributions the paper makes are clearly highlighted. At present, new contributions seem to be scattered throughout the text, and the last paragraph of the conclusions only provides a short comment about them. Some sections, e.g. section 3 which is based on ref. [32] have review-character, while for others (e.g. sections 4-5) it is unclear if their intent is literature review or the introduction of novel material.

In addition, each section currently starts with the problem setup, proceeds with a series of formal calculations and approximations which are introduced scattered throughout the text (e.g. lines 145, 152-153, 158-160 and 169-170 for section 2) and then eventually arrives at a result. The author should reorganise the text so that the result and the assumptions and approximations under which it holds are clearly visible and separated from the actual derivation.

Author Response

 Manuscript ID: Entropy-405128
Reviewer 2
Response (January 16, 2019)

Dear Sir/Lady.

Thanks for your comments.

I have taken into account all your criticisms. I have also taken into account other criticisms.

 Accordingly, I have carried out    two complementary    corrections    throughout  the whole   revised version.

  1.- -In order that    what  is literature review  be clearly distinguished from     the introduction of novel material, I have   modified almost completely Section 8 (Conclusions and discussion). In short, Section 8 summarizes the   contents of the article and distinguishes    review material from   new material, along with some comparative discussions.  

2.- On the other hand, along the whole text, succesive  assumptions, simplifications and approximations are indicated separately. Successive results are explicitly indicated, for  compactness,    within  the main text.
 Thus, without altering the structure, ordering,  construction and equations of the work,   I have   introduced   throughout the whole text the  words “assumption”, “approximation”, “simplification” and “result”, as they become suitable. In this way,    the results and the assumptions, approximations and simplification under which the former  hold are clearly visible and separated from the actual derivations. This improvement in presentation is commented briefly  in two  lines at the end of the Introduction.

And, in so doing, I have also revised the whole manuscript, as an attempt  to improve it.

 Other corrections:

3.- Several paragraphs in the Introduction have been modified.

4. A new correction has been performed, embodied in the new Eq (81) . And a new equation (Eq. (106)) has been added, in order to clarify the justification of a result in Section 7.3.

5) References have been reordered, as commented above, and two new references have been added, for completeness, namely the new references 16 and 56.